



# Delivery of halogenated very short-lived substances from the West Indian Ocean to the stratosphere during Asian summer monsoon

Alina Fiehn[1,2], Birgit Quack[2], Helmke Hepach[2,*], Steffen Fuhlbrügge[2], Susann Tegtmeier[2], Matthew Toohey[2], Elliot Atlas[3], Kirstin Krüger[1]

[1]*Meteorology and Oceanography Section, Department of Geosciences, University of Oslo, Oslo, Norway*
[2] *GEOMAR Helmholtz Centre for Ocean Research Kiel, Kiel, Germany*
[3] *Rosenstiel School of Marine and Atmospheric Science, University of Miami, Miami, USA*
[*] *now at: Environment Department, University of York, York, United Kingdom*

Alina Fiehn: alina.fiehn@geo.uio.no
Birgit Quack: bquack@geomar.de
Helmke Hepach: hhepach@geomar.de
Steffen Fuhlbrügge: sfuhlbruegge@geomar.de
Susann Tegtmeier: stegtmeier@geomar.de
Matthew Toohey: mtoohey@geomar.de
Elliot Atlas: eatlas@rsmas.miami.edu
Kirstin Krüger: kkrueger@geo.uio.no (corresponding author)



**Abstract**

Halogenated very short-lived substances (VSLS) are naturally produced in the ocean and emitted to the atmosphere. When transported to the stratosphere, these compounds can have a significant influence on the ozone layer and climate. During a research cruise on RV *Sonne* in the subtropical and tropical West Indian Ocean in July and August 2014, we measured the VSLS,

methyl iodide ($CH_3I$) and for the first time bromoform ($CHBr_3$) and dibromomethane ($CH_2Br_2$), in surface seawater and the marine atmosphere to derive their emission strengths. Using the Lagrangian transport model Flexpart with ERA-Interim meteorological fields, we calculated the direct contribution of observed VSLS emissions to the stratospheric halogen burden during Asian summer monsoon. Furthermore, we compare the in situ calculations with the interannual

variability of transport from a larger area of the West Indian Ocean surface to the stratosphere for July 2000-2015. We found that the West Indian Ocean is a strong source region for $CHBr_3$ ($910 \, pmol \, m^{-2} \, h^{-1}$), very strong for $CH_2Br_2$ ($930 \, pmol \, m^{-2} \, h^{-1}$), and average for $CH_3I$ ($460 \, pmol \, m^{-2} \, h^{-1}$). The atmospheric transport from the tropical West Indian Ocean surface to the stratosphere experiences two main pathways. On very short timescales, especially relevant for the

shortest-lived compound $CH_3I$ (3.5 days lifetime), convection above the Indian Ocean lifts oceanic air masses and VSLS towards the tropopause. On a longer timescale, the Asian summer monsoon circulation transports oceanic VSLS towards India and Bay of Bengal, where they are lifted with the monsoon convection and reach stratospheric levels in the southeastern part of the Asian monsoon anticyclone. This transport pathway is more important for the longer-lived

brominated compounds (17 and 150 days lifetime for $CHBr_3$ and $CH_2Br_2$). The entrainment of $CHBr_3$ and $CH_3I$ from the West Indian Ocean to the stratosphere during Asian summer monsoon is less than from previous cruises in the tropical West Pacific Ocean during boreal autumn/early winter, but higher than from the tropical Atlantic during boreal summer. In contrast, the projected $CH_2Br_2$ entrainment was very high because of the high emissions during the West Indian Ocean

cruise. The 16-year July time series shows highest interannual variability for the short-lived $CH_3I$ and lowest for the long-lived $CH_2Br_2$. During this time period, a small increase of VSLS entrainment from the West Indian Ocean through the Asian monsoon to the stratosphere is found. Overall, this study confirms that the subtropical and tropical West Indian Ocean is an important source region of halogenated VSLS, especially $CH_2Br_2$, to the troposphere and stratosphere

during the Asian summer monsoon.



# 1 Introduction

Natural halogenated volatile organic compounds in the ocean originate from chemical and biological sources like phytoplankton and macro algae (Carpenter et al., 1999;Quack and Wallace, 2003;Moore and Zafiriou, 1994;Hughes et al., 2011). When emitted to the atmosphere, the halogenated very short-lived substances (VSLS) have atmospheric lifetimes of less than half a year (Law et al., 2006). Current estimates of tropical tropospheric lifetimes are 3.5, 17, and 150 days for methyl iodide ($CH_3I$), bromoform ($CHBr_3$), and dibromomethane ($CH_2Br_2$), respectively (Carpenter et al., 2014). VSLS can be transported to the stratosphere by tropical deep convection, where they contribute to the halogen burden, take part in ozone depletion and thus impact the climate (Solomon et al., 1994;Dvortsov et al., 1999;Hossaini et al., 2015).

$CHBr_3$ is an important biogenic VSLS due to its large oceanic emissions and because it carries three bromine atoms per molecule into the atmosphere (Quack and Wallace, 2003;Hossaini et al., 2012). $CH_2Br_2$ has a longer lifetime than $CHBr_3$, and thus a higher potential for stratospheric entrainment. $CH_3I$ is an important carrier of organic iodine from the ocean to the atmosphere and the most abundant organic iodine compound in the atmosphere (Manley et al., 1992;Moore and Groszko, 1999;Yokouchi et al., 2008). Despite its very short atmospheric lifetime, it can deliver iodine to the stratosphere in tropical regions (Solomon et al., 1994;Tegtmeier et al., 2013). Ship-based observations showed that bromocarbon emissions near coasts and in oceanic upwelling regions are generally higher than in the open ocean, because of macro algal growth near coasts (Carpenter et al., 1999) and enhanced primary production in upwelling regions (Quack et al., 2007), while coastal anthropogenic sources also need to be considered (Quack and Wallace, 2003;Fuhlbrügge et al., 2016b). Measurements of VSLS in the global oceans are sparse and the data shows a large variability. Thus, attempts at creating observation based global emission estimates and climatologies (bottom-up approach) (Quack and Wallace, 2003;Butler et al., 2007;Palmer and Reason, 2009;Ziska et al., 2013), modeling the global distribution of halogenated VSLS emissions from atmospheric abundances (the top-down approach) (Warwick et al., 2006;Liang et al., 2010;Ordóñez et al., 2012), as well as biogeochemical modeling of oceanic concentrations (Hense and Quack, 2009;Stemmler et al., 2014) are subject to large uncertainties (Carpenter et al., 2014). Global modeled top-down estimates (Warwick et al., 2006;Liang et al., 2010;Ordóñez et al., 2012) yield higher emissions



than bottom-up estimates (Ziska et al., 2013;Stemmler et al., 2014, 2015), which may indicate the importance of localized emission hot spots underrepresented in current bottom-up estimates.

The amount of oceanic bromine from VSLS entrained into the stratosphere is estimated to be 2-8 ppt, which is 10-40 % of the currently observed stratospheric bromine loading (Dorf et al., 2006;Carpenter et al., 2014). This wide range results mainly from uncertainties in tropospheric degradation and removal, transport processes, and especially from the spatial and temporal emission variability of halogenated VSLS (Carpenter et al., 2014;Hossaini et al., 2016). Analyzing the time period 1993-2012, Hossaini et al. (2016) found no clear long-term transport-driven trend in the stratospheric injection of oceanic bromine sources during a multi-model intercomparison.

Transport processes strongly impact stratospheric injections of VSLS, because their lifetimes are comparable to tropospheric transport timescales from the ocean to the stratosphere. The main entrance region of tropospheric air into the stratosphere is above the tropical West Pacific. Another active region lies above the Asian monsoon region during boreal summer (Newell and Gould-Stewart, 1981), when the Asian monsoon circulation provides an efficient transport pathway from the atmospheric boundary layer to the lower stratosphere (Park et al., 2009;Randel et al., 2010). Above India and the Bay of Bengal, convection lifts boundary layer air rapidly into the upper troposphere (Park et al., 2009;Lawrence and Lelieveld, 2010). As a response to the persistent deep convection, an anticyclone forms in the upper troposphere and lower stratosphere above Central, South and East Asia (Hoskins and Rodwell, 1995). This so-called Asian monsoon anticyclone confines the air masses that have been lifted to this level within the anticyclonic circulation (Park et al., 2007;Randel et al., 2010). For the period 1951-2015, a decreasing trend in rainfall and thus convection has been reported over northeastern India caused by a weakening northward moisture transport over the Bay of Bengal (Latif et al., 2016).

Chemical transport studies in the Asian monsoon region have mostly focused on water vapor entrainment to the stratosphere (Gettelman et al., 2004;James et al., 2008), or the transport of anthropogenic pollution (Park et al., 2009). The chemical composition and source regions for air masses in the Asian monsoon anticyclone have been the topic of more recent studies (Bergman et al., 2013;Vogel et al., 2015;Yan and Bian, 2015). Chen et al. (2012) investigated air mass boundary layer sources and stratospheric entrainment regions based on a climatological domain-filling Lagrangian study in the Asian summer monsoon area. The West Pacific Ocean





and the Bay of Bengal are found to be important source regions, while maximum stratospheric entrainment occurred above the tropical West Indian Ocean.

The Asian monsoon circulation could be an important pathway for the stratospheric entrainment of oceanic VSLS (Hossaini et al., 2016), because the steady southwest monsoon winds in the lower troposphere during boreal summer deliver oceanic air masses from the tropical Indian Ocean towards India and the Bay of Bengal (Lawrence and Lelieveld, 2010), where they are lifted by the monsoon convection and the Asian monsoon anticyclone. However, little is known about the emission strength of VSLS from the Indian Ocean and their transport pathways.

A few measurements in the Bay of Bengal (Yamamoto et al., 2001) and Arabian Sea (Roy et al., 2011) as well as global source estimates suggest that the Indian Ocean might be a considerable source (Liang et al., 2010;Ziska et al., 2013). No bromocarbon data is available for the equatorial and southern Indian Ocean, yet, but $CH_3I$, which has been measured around the Mascarene Plateau, showed high oceanic concentrations (Smythe-Wright et al., 2005). Liang et al. (2014)

use a Chemistry Climate Model for the years 1960 to 2010 and project that the tropical Indian Ocean delivers more bromine to the stratosphere than the tropical Pacific because of its higher atmospheric surface concentrations based on the global top-down emission estimate by Liang et al. (2010).

In this study, we show surface ocean concentrations and atmospheric mixing ratios of the

135 halogenated VSLS $CH_3I$, and for the first time for $CHBr_3$ and $CH_2Br_2$, in the subtropical and tropical West Indian Ocean during Asian summer monsoon. We use the Lagrangian transport model Flexpart to investigate the atmospheric transport pathways of observation-based oceanic VSLS emissions to the stratosphere.

Our questions for this study are: Is the tropical Indian Ocean a source for atmospheric

VSLS? What is the transport pathway from the West Indian Ocean to the stratosphere during the Asian summer monsoon? How much VSLS are delivered from the West Indian Ocean to the stratosphere during Asian summer monsoon? How large is the interannual variability of this VSLS entrainment?

In Sect. 2, we describe the cruise data and the transport model simulations. In Sect. 3, the

145 results from the cruise measurements and trajectory calculations are shown and discussed. Then the spatial and interannual variability of transport is presented in Sect. 4. In Sect. 5, we address uncertainties before summarizing the results and concluding in Sect 6.





## 2 Data and Methods

### 2.1 Observations during the cruise

During two consecutive research cruises in the West Indian Ocean, we observed meteorological, oceanographic, and biogeochemical conditions, including atmospheric mixing ratios and oceanic concentrations of halogenated VSLS. The two cruises on RV *Sonne*, SO234-2 from July 08 to 19, 2014 (Durban, South Africa - Port Louis, Mauritius) and SO235 from July 23 to August 7, 2014 (Port Louis, Mauritius - Malé, Maldives), were conducted within the SPACES (Science Partnerships for the Assessment of Complex Earth System Processes) and OASIS (Organic very short lived Substances and their air sea exchange from the Indian Ocean to the Stratosphere) research projects. Cruise SO234-2 was an international training and capacity building program for students from Germany and South African countries, whereas SO235 was purely scientifically oriented. The cruise tracks covered subtropical waters, coastal and shelf areas, and the tropical open West Indian Ocean, and were designed to cover biologically productive and non-productive regions (Fig. 1). In the following, we will refer to the combined cruises as the "OASIS cruise".

We collected meteorological data from ship based sensors including surface air temperature (SAT), relative humidity, air pressure, wind speed and direction taken every second at about 25 m height on RV *Sonne*. Sea surface temperature (SST) and salinity (SSS) were measured in the ship's hydrographic shaft at 5 m depth. We averaged all parameters to 10 minute intervals for our investigations.

During the cruise, we launched 95 radiosondes and thus obtained high resolution atmospheric profiles of temperature, wind, and humidity. During the first half of the cruise, regular radiosondes were launched at 0 and 12 UTC, and additionally at 6 and 18 UTC during the 48 hour station (June 16 -18, 2014; Fig. 1). During the second half of the cruise, the launches were always performed at standard UTC times (0, 6, 12, 18 UTC), and every three hours during the diurnal stations (June 26 and 28, Aug. 3, 2014). For the regular launches, we used GRAW DFM-09 radiosondes and for the six ozone sonde launches we used DFM-97. The collected radiosonde data was delivered in near real time to the Global Telecommunication System (GTS) to improve meteorological reanalyses (e.g. European Centre for Medium-Range Weather Forecasts (ECMWF) Re-Analysis Interim (ERA-Interim)) and operational forecast models (e.g. opECMWF (operational ECMWF)) in the subtropical and tropical West Indian Ocean.





Trace gas emissions are generally well mixed within the marine atmospheric boundary layer (MABL) on timescales of an hour or less by convection and turbulence (Stull, 1988). We determined the stable layer that defines the top of the MABL with the practical approach described in Seibert et al. (2000). From the radiosonde ascent we computed the vertical gradient of virtual potential temperature, which indicates the stable layer at the top of the MABL with

positive values. A detailed description of our method can be found in Fuhlbrügge et al. (2013).

We collected a total of 213 air samples with a 3-hourly resolution at about 20 m height above sea level. These samples were pressurized to 2 atm in pre-cleaned stainless steel canisters with a metal bellows pump, and they were analyzed within 6 months after the cruise. Details about the analysis, the instrumental precision, the preparation of the samples, and the use of

190 standard gases are described in Schauffler et al. (1999), Montzka et al. (2003), and Fuhlbrügge et al. (2013).

We collected 154 water samples every three hours from the hydrographic shaft of RV *Sonne* at a depth of 5 m. The samples were then analyzed for halogenated compounds using a purge and trap system onboard, attached to a gas chromatograph with electron capture detector.

Analytical reproducibility of 10 % was determined from measuring duplicate water samples. Calibration was performed with a liquid mixed-compound standard prepared in methanol. Details of the procedure can be found in Hepach et al. (2016).

The sea-air flux ($F$) of the VSLS was calculated from the transfer coefficient ($k_\mathrm{w}$) and the concentration gradient ($\Delta c$) according to Eq. (1). The gradient is between the water concentration

($c_\mathrm{w}$) and theoretical equilibrium water concentration ($c_\mathrm{atm}/H$), which is derived from the atmospheric concentration ($c_\mathrm{atm}$). We use Henry's law constants ($H$) of Moore and coworkers (Moore et al., 1995a;Moore et al., 1995b).

$$F = k_\mathrm{w} \cdot \Delta c = k_\mathrm{w} \cdot \left( c_\mathrm{w} - \frac{c_\mathrm{atm}}{H} \right) \qquad (1)$$

Compound-specific transfer coefficients were determined using the air-sea gas exchange parameterization of Nightingale et al. (2000) and applying a Schmidt number ($Sc$) for the

205 different compounds as in Quack and Wallace (2003) (Eq. (2)).

$$k_\mathrm{w} = k \cdot \frac{Sc^{-\frac{1}{2}}}{600} \qquad (2)$$

Nightingale et al. (2000) determined the transfer coefficient ($k$) as a function of the wind speed at 10 m height ($u_{10}$): $k = 0.2\, u_{10}^2 + 0.3\, u_{10}$. This wind speed is derived from a logarithmic



wind profile using the von Kármán constant ($\kappa = 0.41$), the neutral drag coefficient ($C_d$) from Garratt (1977), and the 10 min average of the wind speed ($u(z)$) measured at $z = 25$ m during the cruise (Eq. (3)):

$$u_{10} = u(z) \frac{\kappa \sqrt{C_d}}{\kappa \sqrt{C_d} + \log \frac{z}{10}} \qquad (3)$$

## 2.2 Trajectory calculations

For our trajectory calculations, we use the Lagrangian particle dispersion model Flexpart of the Norwegian Institute for Air Research in the Department of Atmospheric and Climate Research (Stohl et al., 2005), which has been evaluated in previous studies (Stohl et al., 1998;Stohl and Trickl, 1999). The model includes moist convection and turbulence parameterizations in the atmospheric boundary layer and free troposphere (Stohl and Thomson, 1999;Forster et al., 2007). In this study, we employ the most recently released version 9.2 of Flexpart. We use the ECMWF reanalysis product ERA-Interim (Dee et al., 2011) with a horizontal resolution of 1° x 1° and 60 vertical model levels as meteorological input fields, providing air temperature, winds, boundary layer height, specific humidity, as well as convective and large scale precipitation with a 6-hourly temporal resolution. The vertical winds in hybrid coordinates were calculated mass-consistently from spectral data by the pre-processor (Stohl et al., 2005). We record the transport model output every 6 hours.

We ran the Flexpart model with three different setups, which are described in Table 1. These configurations are designated as 1) *OASIS backward* trajectories, 2) *OASIS* (forward trajectories), and 3) *Indian Ocean* (regional forward trajectories).

We calculate *OASIS backward* trajectories from the 12 UTC locations of RV *Sonne* during the cruise. These trajectories are later used to determine the source regions of air masses investigated along the cruise track.

With the *OASIS* setup, we study the transport of oceanic $CHBr_3$, $CH_2Br_2$, and $CH_3I$ emissions from the measurement locations into the stratosphere similar as was carried out in the corresponding study by Tegtmeier et al. (2012). At every position along the cruise track at which emissions were calculated (Section 2.1), we release a mass of the compound equal to a release from 0.0002° x 0.0002° in one hour. The mass is evenly distributed among 10,000 trajectories. During transport, $CHBr_3$ and $CH_2Br_2$ mass is depleted according to atmospheric lifetime profiles from Hossaini et al. (2010) based on Chemistry Transport Model simulations including VSLS




chemistry. CH$_3$I decays applying a uniform vertical lifetime of 3.5 days (Sect. 1). The mass on all trajectories that reach a height of 17 km is summed and assumed to be entrained into the

stratosphere. This threshold height represents the average cold point tropopause (CPT) height during the cruise (see Fig. S2) and also for the whole tropics (Munchak and Pan, 2014). The influence of the entrainment height criteria is further discussed in Sect. 4. For intercomparison with other ocean basins, we employed exactly the same model-setup of transport simulations (including lifetimes) and the same emission calculation method for three previous corresponding

cruises in the tropics: the TransBrom campaign in the West Pacific in 2009 (Introduction to special issue: Krüger and Quack, 2013), the SHIVA campaign in the South China and Sulu Seas in 2011 (Fuhlbrügge et al., 2016a), and the MSM18/3 cruise in the equatorial Atlantic Cold Tongue (Hepach et al., 2015).

The transport calculations based on the measured emissions from OASIS give insight into

the contribution of oceanic emissions to the stratosphere during Asian summer monsoon. However, transport and emissions in the *OASIS* study are localized in space and time, and could thus be very different for different areas and years. In order to investigate the transport from the West Indian Ocean basin to the stratosphere and its interannual variability under the influence of the Asian summer monsoon circulation (*Indian Ocean* setup), we calculate trajectories from a

large region of the tropical West Indian Ocean surface for the years 2000-2015. Trajectories are uniformly started within the release area (50°E-80°E, 20°S-10°N), covering the tropical West Indian Ocean, once every day during July 2000-2015. The run time is set to 3 months, which covers the period July to October. We then calculate the fraction (*q*) of each *VSLS tracer* that reaches the stratosphere during the transit time (*tt*) assuming an exponential decay of the tracer

(Eq. 4) according to the tropical tropospheric lifetimes (*lt*) of 17, 150, and 3.5 days for CHBr$_3$, CH$_2$Br$_2$, and CH$_3$I, respectively (Carpenter et al., 2014).

$$q = e^{-\frac{tt}{lt}} \qquad (4)$$

We use the term "VSLS tracer" to distinguish from the calculations used in the *OASIS* setup, where actual VSLS emissions experience decay according to a vertical lifetime profile (uniform for CH$_3$I). This *Indian Ocean* setup provides information on the preferred pathways

from the West Indian Ocean to the stratosphere for different transport timescales and on their interannual variability. This variability is quantified by the coefficient of variation (CV), which is defined as the ratio of the standard deviation to the mean entrainment. The correlations of the





interannual variations between different regions of stratospheric entrainment are given by the correlation coefficient (*r*) by Pearson (1895).


## 3 The Indian Ocean cruise: OASIS

### 3.1 Atmospheric circulation

SST and SAT during the OASIS cruise generally increase from south towards the equator (Fig. 2a). The SST is on average 1.5 °C higher than the SAT, which benefits convection.

Minimum SSTs of 18 °C were measured from July 14 to 17, 2014 in the open subtropical Indian Ocean (30° S, 59° E) and maximum SSTs of 29 °C were measured around the equator.

The overall mean wind speed was 8.1 m s$^{-1}$, with lower wind speeds in the subtropics and close to the equator (5 m s$^{-1}$), and higher wind speeds (up to 15 m s$^{-1}$) in the trade wind region (July 23 to August 5, 20° - 5° S) (Fig. 2b). The mean wind direction during the cruise was

southeast. While the wind direction showed large variability in the subtropics, southeasterly trade winds dominated between Mauritius and the equator. North of the equator the wind direction changed to westerly winds. Our in situ ship wind measurements deviate from the mean July wind field from ERA-Interim during the first part of the cruise south of Mauritius (Fig. 1a) due to the influence of a developing low pressure system (not shown). The steady trade winds during the

second part of the cruise are well reflected in the July mean wind field from ERA-Interim.

Air masses sampled during the cruise originate mainly from the open ocean (Fig. 3a). Trajectories started between South Africa and Mauritius generally come from the south. An influence of terrestrial sources is possible close to South Africa and Madagascar. From Mauritius to the Maldives, the trajectories originate from the southeast open Indian Ocean. The analysis of

air samples reveal no recent fresh anthropogenic input, indicated by the very low levels of short-lived trace gas contaminants, e.g. butane, in this region (not shown).

### 3.2 VSLS observations and oceanic emissions

CHBr$_3$, CH$_2$Br$_2$, and CH$_3$I surface ocean concentrations, atmospheric mixing ratios, and

emissions for the OASIS cruise are plotted as time series in Fig. 2c-e, and are summarized in Table 2.

CHBr$_3$ concentrations in the surface ocean range from 1.3 to 33.4 pmol L$^{-1}$ with an average over all measurements of 8.4 ± 14.2 (1σ) pmol L$^{-1}$. The standard deviation (σ) is used as



a measure of the variability in the measurements during the cruise. We measured higher water

concentrations of >10 pmol L$^{-1}$ close to coasts and shelf regions and in the open Indian Ocean

between 5° and 10° S (July 27- Aug. 2). Oceanic concentrations of $CH_2Br_2$ are lower, with a

mean of $6.7 \pm 12.6$ pmol L$^{-1}$, but show a similar pattern to $CHBr_3$ concentrations. High

concentrations were measured southeast of Madagascar, when we passed the southern stretch of

the East Madagascar Current. Oceanic upwelling occurs along the eddy-rich, shallow region

south of Madagascar, which leads to locally enhanced phytoplankton growth (Quartly et al.,

2006), and possibly upwelling of elevated $CH_2Br_2$ concentrations from the deeper ocean could

have occurred similar as was observed for the equatorial upwelling in the Atlantic (Hepach et al.,

2015). $CH_3I$ oceanic concentrations range from 0.2 to 16.4 pmol L$^{-1}$ with a mean of

$3.4 \pm 3.1$ pmol L$^{-1}$. They were elevated (5-12 pmol L$^{-1}$) during the last part of the cruise (Aug. 3-6,

2014) around the equator. In the region of the Mascarene Plateau, to the west of our cruise,

Smythe-Wright et al. (2005) detected much higher $CH_3I$ concentrations between 20 and

40 pmol L$^{-1}$ during June-July 2002.

Atmospheric mixing ratios $CHBr_3$ during the OASIS cruise (Fig. 2d, Table 2) show an

overall mean of $1.20 \pm 0.35$ ppt. Elevated mixing ratios of >2 ppt are found in three locations:

south of Madagascar, in Port Louis, and close to the British Indian Ocean Territory. The first two

have probably terrestrial or coastal sources, because they do not coincide with high oceanic

$CHBr_3$ concentrations but backward trajectories pass land. Close to the British Indian Ocean

Territory, oceanic concentrations and atmospheric mixing ratios are elevated, which suggests a

local oceanic source. Atmospheric mixing ratios of $CH_2Br_2$ vary little around the average of

0.91 ppt and show a similar pattern as the $CHBr_3$ mixing ratios. $CH_3I$ ($0.84 \pm 0.12$ ppt) mixing

ratios show pronounced variations and surpass 1 ppt in some locations. These atmospheric

mixing ratios above the open ocean are much lower than the average of 12 pptv Smythe-Wright

et al. (2005) reported around the Mascarene Plateau.

We calculated oceanic emissions from the synchronized measurements of surface water

concentration and atmospheric mixing ratio as described in Section 2.1 (Fig. 2e and Fig 1). High

emissions are caused by high oceanic concentrations, high wind speeds, or a combination of both.

The OASIS emission strength of $CHBr_3$ ranges from -100 to 9630 pmol m$^{-2}$ hr$^{-1}$ with high mean

emissions of $910 \pm 1,160$ pmol m$^{-2}$ hr$^{-1}$, caused by moderate water concentrations and relatively

high wind speeds. We derive the highest emissions south of Madagascar and in the trade wind

regime from 5° S to 10° S above the open ocean upwelling region of the Seychelles-Chagos-



thermocline ridge (Schott et al., 2009), where we also observed enhanced phytoplankton growth (not shown here). $CH_2Br_2$ emissions (with an overall mean of $930 \pm 2,000$ pmol m$^{-2}$ hr$^{-1}$) were by far highest south of Madagascar with a single maximum of up to 20,000 pmol m$^{-2}$ hr$^{-1}$. Here, we experienced very high oceanic concentrations and high wind speeds due to the passage of a low

pressure system south of the ship track during July 11-17, 2014. $CH_3I$ emissions ($460 \pm 430$ pmol m$^{-2}$ hr$^{-1}$) had a pronounced maximum of 2,090 pmol m$^{-2}$ hr$^{-1}$ around 10° S and 70° E (July 31- Aug. 1), in accordance with high wind speeds and oceanic concentrations being elevated close to the above mentioned open ocean upwelling observed between 5° and 10°S.

During the first part of the cruise, we recorded low mean atmospheric mixing ratios of

$CHBr_3$ and $CH_2Br_2$, despite high local oceanic concentrations and emissions especially south of Madagascar. In connection with a high and well ventilated MABL (Fig. S2), this indicates that the strong sources south of Madagascar are highly localized. The occasional enhancement of the brominated VSLS in some air samples underlines the patchiness of the sources in this region. During the second part of the cruise, the atmospheric mixing ratios of $CHBr_3$ and $CH_2Br_2$

increased from south to north and in the direction of the wind maximizing close to the equator (Fig. 2d). The emissions were high between Mauritius and the equator (Fig. 2e). This suggests that the air around the equator was enriched by the advection of the oceanic emissions with the trade winds from south to north. We assume that the bromocarbons accumulate because of the steady wind directions and the suppression of mixing into the free troposphere by the top of the

MABL and the trade inversion layer (Fig. S2, July 27- Aug. 2) acting as strong transport barriers for VSLS as was observed for the Peruvian upwelling (Fuhlbrügge et al., 2016a).

### 3.3    Comparison of OASIS VSLS emissions with other oceanic regions

Average emissions of the three VSLS from OASIS and other tropical cruises and modeling

studies are summarized in Table 3. We compare with cruises and open ocean estimates, since OASIS mainly covered open ocean regions and only small coastal areas close to Madagascar, the British Indian Ocean Territory and the Maldives.

The average $CHBr_3$ emission during the OASIS campaign (910 pmol m$^{-2}$ hr$^{-1}$) was higher than during most campaigns in tropical regions: 1.5 times higher than during TransBrom in the

subtropical and tropical West Pacific (Tegtmeier et al., 2012), 1.2 times higher than during DRIVE in the tropical North East Atlantic (Hepach et al., 2014), and 1.5 times higher than during MSM18/3 in the Atlantic equatorial upwelling (Hepach et al., 2015). Only the SHIVA campaign



in the South China and Sulu Seas yielded higher $CHBr_3$ emissions of 1,486 pmol m$^{-2}$ hr$^{-1}$ because of very high oceanic concentrations close to the coast (Fuhlbrügge et al., 2016b). The global open

ocean estimate by Quack and Wallace (2003) is one-third lower than our measured values in the West Indian Ocean. The bottom-up emission climatology by Ziska et al. (2013) estimates lower values for the Indian Ocean, based on measurements from other oceanic basins due to a lack of available Indian Ocean in situ measurements. With their top-down approach, Warwick et al. (2006), Liang et al. (2010), and Ordóñez et al. (2012) calculated $CHBr_3$ emissions in the range of

580-956 pmol m$^{-2}$ hr$^{-1}$ for the tropical ocean. Stemmler et al. (2014) modeled very low $CHBr_3$ emissions around 200 pmol m$^{-2}$ hr$^{-1}$ for the equatorial Indian Ocean with their biogeochemical ocean model.

Average $CH_2Br_2$ emissions from the OASIS cruise (930 pmol m$^{-2}$ hr$^{-1}$) are 2-6 times higher than average cruise emissions listed in Table 3: TransBrom, DRIVE, MSM18/3, SHIVA,

and M91. This is caused by the generally high oceanic concentrations during OASIS, with highest values south of Madagascar. The mean emissions from the West Indian Ocean are also much higher than the tropical ocean estimate from Butler et al. (2007), and the global open ocean estimate from Yokouchi et al. (2008) and Carpenter et al. (2009). The top-down model approach by Liang et al. (2010) yielded the lowest emissions of only 81 pmol m$^{-2}$ hr$^{-1}$. The Ziska et al.

(2013) climatology shows maximum equatorial Indian Ocean $CH_2Br_2$ emission values around 500 pmol m$^{-2}$ hr$^{-1}$.

The average $CH_3I$ emissions during OASIS (460 pmol m$^{-2}$ hr$^{-1}$) were in the range of previously observed and estimated values from 254 to 625 pmol m$^{-2}$ hr$^{-1}$ (Table 3). Only for the highly productive Peruvian Upwelling, Hepach et al. (2016) calculated much higher emissions of

954 pmol m$^{-2}$ hr$^{-1}$. The coupled ocean-atmosphere model of Bell et al. (2002) produced average global emissions of 670 pmol m$^{-2}$ hr$^{-1}$, while Stemmler et al. (2013) modeled $CH_3I$ emissions around 500 pmol m$^{-2}$ hr$^{-1}$ for the tropical Atlantic with their biogeochemical ocean model. The Ziska et al. (2013) climatology shows Indian Ocean $CH_3I$ emissions around 500 pmol m$^{-2}$ hr$^{-1}$.

**3.4     VSLS entrainment to the stratosphere during OASIS**

The *OASIS* forward trajectories released at the locations of the VSLS measurements show the transport pathway of the air masses from their sample points along the cruise track (Fig. 3b). The mean of all 10,000 trajectories from each release can be grouped into four regimes according to transport direction: Westerlies, Transition, Monsoon Circulation, and Local Convection. The air





masses in the Westerlies regime are transported to the Southeast Indian Ocean and the air masses from the Transition regime propagate towards Madagascar and Africa. Flexpart calculations reveal that both transport regimes lift air masses up to a mean height of about 5.3 km after one month (not shown here). The trajectories of the Monsoon Circulation regime first travel with the southeasterly trade winds and then with the southwesterly monsoon winds. The trajectories stay

relatively close to the ocean surface (below 3 km) until they reach the Bay of Bengal, where they are rapidly lifted to the upper troposphere. On average they reach a height of 7.9 km after one month, which reveals that this is the regime with most convection. The trajectories of the Local Convection regime mainly experience rapid uplift around the equator. After one month this group has reached a mean height of 7.2 km.

The absolute entrainment of oceanic VSLS to the stratosphere depends on the emission strength as well as the *transport efficiency* (Fig. 4). This efficiency is defined as the ratio of entrained to emitted VSLS. It depends on the *transit time*, defined as the time an air parcel needs to be transported from the ocean surface to 17 km height, and the lifetime of the compound. For stratospheric entrainment the transit time must be in the order of the lifetime of a compound or

shorter. If the transit time is considerably larger than the lifetime, most of the compound has decayed before reaching the stratosphere. In the following, we will use the expressions *VSLS transit time*, which is the transit time including loss processes of the VSLS in the atmosphere during the transport, and *transit half-life*, which is the time after which half of the total entrained compound has reached 17 km. We also calculated the relative emission and entrainment by

regime. Table 4 displays the absolute and relative emissions and entrainment, the transport efficiency, and the transit half-life for the whole cruise and the four regimes.

    The mean sea surface release of $CHBr_3$ in Flexpart is 0.43 µmol (on 0.0002° x 0.0002° hr$^{-1}$) during the cruise and the mean entrainment to the stratosphere is 5.5 nmol resulting in a mean transport efficiency of 1.3 %. $CH_2Br_2$ has a higher transport efficiency of 6.4 % with

mean emissions of 0.43 µmol (on 0.0002° x 0.0002° hr$^{-1}$), and very high stratospheric entrainment of 23.6 nmol. $CH_3I$ has a low transport efficiency of 0.3 % with mean emissions of 0.22 µmol (on 0.0002° x 0.0002° hr$^{-1}$) and stratospheric entrainment of 0.7 nmol.

    The four transport regimes show different transport efficiencies of $CHBr_3$, $CH_2Br_2$, and $CH_3I$ to the stratosphere. The two most efficient regimes, transporting $CHBr_3$ and $CH_3I$ to the

stratosphere during the OASIS cruise, were the Monsoon Circulation and the Local Convection regime.





The transport efficiency for all three compounds is highest in the Local Convection regime (CHBr$_3$ ~3 %, CH$_2$Br$_2$ ~9 %, and CH$_3$I ~1 %), because this regime has the shortest transit half-life for all three VSLS. For CH$_3$I, the compound with the shortest lifetime, the fast transport plays the largest role, and thus this regime is by far the most efficient.

For CHBr$_3$, the regime with most absolute and relative stratospheric entrainment (11 nmol, 57 %) is the Monsoon Circulation regime, because of the high emissions in the source region and the high transport efficiency. Although the CHBr$_3$ emissions are as high in the Westerlies regime, the entrainment is small (2 nmol, 9 %), because of a low transport efficiency due to slow transport visible in the long transit half-life. The Local Convection regime has the highest transport efficiency, but emissions were low, resulting in less entrainment (4 nmol, 23 %) than in the Monsoon Circulation regime. The absolute entrainment of CH$_2$Br$_2$ strongly depends on the strength of emission, because the transport efficiency is relatively similar for all transport regimes due to the long lifetime of the compound. Most entrained CH$_2$Br$_2$ comes from the Westerlies regime (29 nmol, 35 %), where sources especially south of Madagascar were extremely strong. Although these emissions occur in the subtropics, they reach 17 km mainly in the tropics (Fig. S3). The transport efficiency of 4 % still allows a large amount of 345 nmol CH$_2$Br$_2$ to enter the stratosphere from the maximum emissions at 23 UTC on July 12, 2014 (Fig. 4). CH$_3$I absolute entrainment (2.8 nmol, 79 %) is highest in the Local Convection regime, because of both highest emissions and highest transport efficiency (Table 4).

## 3.5 Comparison of VSLS entrainment to the stratosphere with other oceanic regions

A comparison of the subtropical and tropical Indian Ocean contribution to the stratosphere with other tropical ocean regions, applying the same emission calculation and model-setup (Sect. 2.2) for CHBr$_3$ is shown in Table 5. Though the Western Pacific TransBrom cruise had lower bromoform emission rates compared to OASIS, stratospheric entrainment was greater for the Western Pacific region compared to the Indian Ocean. This difference was caused by a higher transport efficiency of 4.4 % in the West Pacific influenced by tropical cyclone activity in October 2009 (Krüger and Quack, 2013). Tegtmeier et al. (2012) obtained a higher transport efficiency of 5 % for TransBrom using a previous Flexpart model (version 8.0). During the SHIVA campaign in the South China Sea, high oceanic concentrations of bromoform produced mean emission rates that were higher than during OASIS. The SHIVA calculations show even



higher transport efficiencies of 7.9 %, which lead to an entrainment of 48.4 nmol $CHBr_3$
(Table 5), because of the strong convective activity in that region during the time (Fuhlbrügge et
al., 2016b). The MSM18/3 cruise in the equatorial Atlantic (Hepach et al., 2015) has the smallest
emissions, entrainment, and a transport efficiency of 0.8 % (Table 5). Overall, the comparison
indicates that more $CHBr_3$ was entrained to the stratosphere from the tropical West Pacific than
from the tropical West Indian Ocean during Asian summer monsoon, using available in situ
emissions and 6-hourly meteorological fields. This is in contrast to the study by Liang et al.
(2014), who projected with a chemistry climate model climatology that emissions from the
tropical Indian Ocean deliver more brominated VSLS into the stratosphere than tropical West
Pacific emissions.

$CH_2Br_2$ entrainment to the stratosphere for the TransBrom ship campaign was ~8 nmol with
transport efficiencies of 15 % (Tegtmeier et al., 2012). This is much higher than the Indian Ocean
transport efficiency of 6.4 %, but the absolute entrainment of 23.6 nmol $CH_2Br_2$ we calculated for
the OASIS cruise (Table 4) is much higher than during TransBrom, because of the very strong
$CH_2Br_2$ emissions during OASIS.

Tegtmeier et al. (2013) investigated $CH_3I$ entrainment to the stratosphere for three tropical
ship campaigns: SHIVA and TransBrom in the tropical West Pacific, and DRIVE in the tropical
North East Atlantic. They used a $CH_3I$ lifetime profile between 2-3 days. The transport
efficiencies were 4 %, 1 %, and 0.1 %, respectively. The OASIS Indian Ocean mean transport
efficiency for $CH_3I$ (0.3 %, Table 4), applying a uniform lifetime profile of 3.5 days, is lower
than in the West Pacific, but higher than in the Atlantic.

Uncertainties of VSLS emissions and modeling their transport to the stratosphere will be
further discussed in Sect. 5.

## 4        General transport from West Indian Ocean to the stratosphere
## 4.1     Spatial variability of stratospheric entrainment
We calculate the entrainment at 17 km for $CHBr_3$, $CH_2Br_2$, and $CH_3I$ tracers by weighting the
trajectories from the West Indian Ocean release region for July 2000-2015 with the transit-time-
dependent atmospheric decay plotted in Fig. 5. A summary of transport efficiency, transit half-
life, and entrainment correlations for all three VSLS can be found in Table 6.



The distribution of VSLS transit times shows that the shorter the lifetime of a compound, the more important is the transport on short timescales (Fig. 5). For $CHBr_3$, $CH_2Br_2$, and $CH_3I$ tracers the transit half-lifes are 8.5, 27.2, and 1.9 days, respectively (Table 6). For the two bromocarbons, the transit time distribution shows two maxima, one for the 0-2 days bin, and the second between 4-10 days for $CHBr_3$ and 6-12 days for $CH_2Br_2$. $CH_3I$ tracer entrainment occurs

mainly on timescales up to 2 days (Fig. 5).

The stratospheric entrainment regions during Asian summer monsoon between 2000 and 2015 are displayed at the locations where the trajectories first reach 17 km (Fig. 6). The VSLS tracers show two main entrainment regions. Enhanced entrainment occurs above Bay of Bengal and northern India in the southeastern part of the Asian monsoon anticyclone, and is connected to

the Monsoon Circulation transport regime (Sect. 3.4). The second entrainment region is above the tropical West Indian Ocean, and belongs to the Local Convection regime. We define these two regions to enclose the core entrainment and to be evenly sized (colored boxes in Fig. 6).

The larger West Indian Ocean release area and longer time series analysis (Table 6) confirms the results of our OASIS analysis (Table 4). The longer-lived VSLS tracers ($CHBr_3$ and

$CH_2Br_2$) are mainly entrained through the Monsoon Circulation regime, while the Local Convection regime is more important for the shortest-lived tracer ($CH_3I$).

Chen et al. (2012) also identified these two stratospheric entrainment regions, analyzing the air transport from the atmospheric boundary layer to the tropopause layer in the Asian Summer monsoon region for a 9 year climatology. Additionally, they registered entrainment over

the West Pacific Ocean, but the Local Convection entrainment above the central Indian Ocean was by far the strongest. Similar to our VSLS transit times, the study of Chen et al. (2012) found very short transport timescales of 0-1 days in the equatorial West Indian Ocean, while transit times above the Bay of Bengal and northern India were between 3 and 9 days.

**4.2     Interannual variability of stratospheric entrainment**

The time series of stratospheric entrainment from the West Indian Ocean to the stratosphere shows interannual variability for all three VSLS tracers (Fig. 7). Overall, July 2014 revealed high entrainment for $CHBr_3$ and $CH_2Br_2$ tracers and low entrainment for $CH_3I$ tracer. The coefficient of variation (CV) for total entrainment is 0.13, 0.09, and 0.21 for $CHBr_3$, $CH_2Br_2$ and $CH_3I$,

respectively. Thus, the shortest-lived compound $CH_3I$ has the strongest interannual variation, the longest-lived $CH_2Br_2$ the weakest variation.



In order to analyze which transport regime has a stronger influence on the total entrainment variability, we correlated the interannual entrainment time series of total entrainment with the entrainment in the Monsoon Circulation and Local Convection regimes (Table 6). Interannual

variability of $CHBr_3$ and $CH_2Br_2$ tracer entrainment results mainly from variability in the Monsoon Circulation regime (r = 0.54 and r = 0.56, respectively). In contrast, the interannual variability of $CH_3I$ tracer entrainment is highly correlated with the Local Convection regime variability (r = 0.87). The high variability of total $CH_3I$ entrainment (CV = 0.21) implies that interannual variation in convection is larger than in the monsoon circulation. The interannual

time series of Monsoon Circulation and Local Convection regime reveal a weak inverse correlation for all three compounds, suggesting that more entrainment in one regime is related to less entrainment in the other (Fig. 7).

The interannual time series of total VSLS tracer entrainment displays a small increase over time. This increase is independent of the chosen entrainment height (between 13 km and 19 km,

Fig. S4), and is visible in the analysis for all three tracers. The increase is strongest for $CHBr_3$ and weaker for the other two compounds. It arises mainly from an increase of entrainment in the Monsoon Circulation regime (Fig. 7). Analyzing changes of rainfall revealed an increase in precipitation over northeastern India for the time interval of our transport study (Latif et al., 2016;Preethi et al., 2016). This indicates an increase of convection in our Monsoon Circulation

regime over the years from 2000 to 2015, which can explain the increase in stratospheric entrainment. However, for the long time period from the 1950s to the 2010s the same authors found a decrease of precipitation over the above mentioned area, potentially impacting the VSLS entrainment to the stratosphere.

In a follow-up study we will investigate the influence of the seasonal cycle of the Asian

Monsoon circulation and interannual influences through atmospheric circulation patterns on the West Indian Ocean VSLS entrainment to the stratosphere in more detail.

## 5      Uncertainties in the analysis

This study confirms that the subtropical and tropical West Indian Ocean is a source region of

oceanic halogenated VSLS to the stratosphere during the Asian summer monsoon. The amount of VSLS entrained depends on the emission strength, the lifetime of the compound, and the transport of trajectories in the regime, which have been quantified in this study.



However, uncertainties of this study are present in various aspects of the analysis. The uncertainties result from the calculation of VSLS emissions, the Flexpart transport using ERA-Interim reanalysis fields, and the definition of entrainment to the stratosphere.

The calculation of VSLS emissions from the concentration gradient between the ocean at 5 m depth and the atmosphere at 20 m height is subject to measurement uncertainties and a possible different concentration gradient directly at the air-sea interface. Also the applied wind-speed-based parameterization for air-sea flux, which represents a reasonable mean of the published parameterizations, is uncertain by more than a factor of two (Lennartz et al., 2015). Both factors may lead to a systematic flux under- or overestimation in our study.

A vital part of this study is the meteorological reanalysis data ERA-Interim and the Flexpart model for determining the VSLS transport. With delivery of our radiosonde launches to the GTS we have improved the data coverage over the Indian Ocean for the time of our study, and thus the quality of meteorological reanalysis. Indeed, horizontal wind speed and direction from ship sensors and sondes agree well with the ERA-Interim data (Fig. S1). As the scale of tropical convection is below the state of the art grid-scale of global atmospheric models it is not sufficiently resolved and must be parameterized. The Lagrangian model Flexpart uses a convection scheme, described and evaluated by Forster et al. (2007), to account for vertical transport. Using Flexpart trajectories with ERA-Interim reanalysis, Fuhlbrügge et al. (2016b) were able to simulate VSLS mixing ratios from the surface to the free troposphere up to 11 km above the tropical West Pacific in very good agreement with corresponding aircraft measurements applying a simple source-loss approach. Tegtmeier et al. (2013) showed that Flexpart distribution of oceanic $CH_3I$ in the tropics agrees well with adjacent upper tropospheric and lower stratospheric aircraft measurements, thus increasing our confidence in the Flexpart convection scheme and ERA-Interim velocities. Testing different Flexpart model versions (8.0 and 9.2) for stratospheric entrainment of $CHBr_3$ (not shown), has revealed only slightly lower stratospheric entrainment of 0.2 % with the more recent model version 9.2 used in this study here.

Another uncertainty in the location and variability of entrained trajectories may result from the definition of stratospheric entrainment (Sect. 2.2). For the tropics, the Cold Point Tropopause (CPT) is commonly used as boundary between the troposphere and the stratosphere (Carpenter et al., 2014). The average measured CPT height during OASIS was 17 km (Fig. S2), but it can be up to 17.6 km high within the Asian monsoon anticyclone during boreal summer season (Munchak and Pan, 2014). To test the sensitivity of our results with regard to the



entrainment height, we analyzed entrained trajectories at several different tropical levels in the upper troposphere/lower stratosphere (UTLS) (13, 15, 17, and 18 km altitude, Fig. S4). As described in Sect. 3.4, we can follow the preferred transport pathways by the migration of maximum density at the intersecting UTLS levels. Analyzing the influence of applying different UTLS entrainment levels reveal an overall good agreement of interannual variability and long-

term changes (Fig. S5).

## 6      Summary and conclusion

During the research cruise OASIS in the subtropical and tropical West Indian Ocean in July and August 2014, we conducted simultaneous measurements of the halogenated very short-lived

substances (VSLS) methyl iodide ($CH_3I$), and for the first time of bromoform ($CHBr_3$) and dibromomethane ($CH_2Br_2$), in surface seawater and the marine atmosphere. Based on these measurements, we calculated high emissions of $CHBr_3$ of $910 \pm 1160$ pmol m$^{-2}$ hr$^{-1}$ caused by high oceanic concentrations south of Madagascar, and moderate concentrations combined with high wind speeds (up to 15 ms$^{-1}$) in the trade wind regime above the open West Indian Ocean.

The average $CHBr_3$ emissions were at the higher end of previously reported values of the tropical oceans. $CH_2Br_2$ emissions of $930 \pm 2,000$ pmol m$^{-2}$ hr$^{-1}$ were especially high also south of Madagascar, and on average higher than reported from cruises in other tropical regions, from global observational and model based climatologies. $CH_3I$ emissions ($460 \pm 430$ pmol m$^{-2}$ hr$^{-1}$) were highest around the equator, but in the range of previously reported emission rates from

subtropical and tropical ocean regions.

      The stratospheric entrainment of these three VSLS from the West Indian Ocean during Asian summer monsoon depends on the strength of emissions and the timescale of the transport to the stratosphere in comparison to the lifetime of the compound. The entrainment of the shortest-lived compound $CH_3I$ (3.5 days) depends mainly on fast transport. The entrainment of

$CH_2Br_2$ strongly depends on the emission strength, because the transport efficiency is relatively similar for all transport regimes due to the long lifetime of the compound (150 days). $CHBr_3$ (17 days lifetime) entrainment is influenced by both oceanic emissions and fast transport.

      During the OASIS cruise we found four transport regimes with different VSLS emission strengths and transport efficiencies. The Monsoon Circulation and the Local Convection regime

were most efficient for VSLS entrainment into the stratosphere. These two have different source





regions, VSLS transit times, stratospheric entrainment regions, and interannual variations summarized in Fig. 8.

In the Monsoon Circulation regime, the oceanic VSLS transport pathway begins south of the equator and follows the near surface winds to India and Bay of Bengal, where monsoon convection rapidly lifts them into the upper troposphere. The VSLS ascend further within the Asian monsoon anticyclone, being entrained to stratospheric levels in its southeastern part. The transport to the stratosphere in this regime is effective for $CHBr_3$ and $CH_2Br_2$ (2 % and 8 % transport efficiency, respectively), but less effective for $CH_3I$ (0.3 %) as its lifetime is shorter than the transport timescale. Absolute $CHBr_3$ entrainment from the OASIS cruise was strongest in the Monsoon Circulation regime because of strong emissions in the source region. The *Indian Ocean* setup showed that it is generally the preferred regime for the entrainment of VSLS with longer lifetimes during boreal summer, because many trajectories follow this transport pathway. Mean transit half-lifes from the West Indian Ocean surface to 17 km height are 6 days for $CHBr_3$ and 13 days for $CH_2Br_2$.

In the Local Convection regime, VSLS are transported upwards by convection above the tropical West Indian Ocean and entrained to the stratosphere in the vicinity of the equator. VSLS transit times are short (0-2 days) and thus we found the highest transport efficiencies for $CHBr_3$, $CH_2Br_2$, and $CH_3I$ in this region (3 %, 9 %, and 1 %). The Local Convection regime is responsible for most of the stratospheric entrainment of $CH_3I$ from the OASIS cruise. The *Indian Ocean* transport study supports this finding.

$CH_2Br_2$ transport efficiency is similar for all regimes of the OASIS cruise, because its lifetime is longer than the transport timescale from ocean to stratosphere in the tropics. Absolute entrainment of $CH_2Br_2$ thus strongly depends on the strength of emissions and these were very high during OASIS, especially south of Madagascar.

In comparison to other corresponding cruises, the Monsoon Circulation and Local Convection regime in the tropical West Indian Ocean show more entrainment of $CHBr_3$ and $CH_3I$ than the tropical Atlantic, but less than the tropical West Pacific Ocean. $CH_2Br_2$ entrainment from the West Indian Ocean was higher than from previous corresponding cruises in other tropical oceans due to the very high emissions.

A 16-year time series (2000-2015) of VSLS tracer entrainment from the West Indian Ocean to the stratosphere through the monsoon circulation during July reveals strongest interannual variability for $CH_3I$, the shortest-lived compound, which seems to be connected to the interannual



variation in convection above the West Indian Ocean. The weakest variations were found for $CH_2Br_2$, the longest-lived compound, whose entrainment hardly depends on the local atmospheric circulation. The time series of entrainment to the stratosphere shows an overall increase for all three compounds, which is likely connected to a reported increase of precipitation and convection over northeastern India during this time period. For $CHBr_3$, whose transport is mostly associated with the changing Asian summer monsoon circulation, the increase is stronger than for the other two compounds.

Overall, the OASIS measurements confirm that during boreal summer the subtropical and tropical West Indian Ocean is an important source for VSLS, especially of $CH_2Br_2$, with pronounced hot spots. This study demonstrates that the VSLS delivery from the West Indian Ocean surface to the stratosphere depends on the regional strength of emissions and the transit time in preferred transport regimes. Changes in the Asian summer monsoon circulation likely impact the VSLS entrainment to the stratosphere.



## Data availability

The underlying data will be available at the open-access library Pangaea (http://www.pangaea.de).

## Author contribution

A. Fiehn, K. Krüger and B. Quack designed the experiments and A. Fiehn carried them out. All
coauthors were involved in the VSLS measurements and analyses taken during the OASIS cruise.
A. Fiehn, K. Krüger and B. Quack prepared the manuscript with contributions from all co-authors.

## Competing interests

The authors declare that they have no conflict of interest.

## Acknowledgements

This study was supported by BMBF grant SONNE 03G0235A. We acknowledge the authorities
of Madagascar, Mauritius, the United Kingdom, and the Maldives for the permissions to work in
their territorial waters. We would also like to thank the captain and crew of RV *Sonne* as well as
the student participants during SPACES for their help and cooperation. We thank the European
Centre for Medium-Range Weather Forecasts (ECMWF) for the provision of ERA-Interim
reanalysis data and the Flexpart development team for the Lagrangian particle dispersion model
used in this publication. Part of the Flexpart simulations was performed on resources provided by
UNINETT Sigma2 - the National Infrastructure for High Performance Computing and Data
Storage in Norway. E. Atlas acknowledges support from NASA Upper Atmosphere Program
Grant #NNX13AH20G.

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





**Table 1: FLEXPART experimental set ups including experiment name, mode, start location and time, runtime, and number of trajectories.**

| Experiment Name | Mode | Start location | Start time | Runtime | Number of trajectories |
|---|---|---|---|---|---|
| *OASIS back* | Backward; air mass | along ship track | 12 UTC, every day during cruise | 10 days | 50 per cruise day |
| *OASIS* | Forward; VSLS | 0.0002° x 0.0002° on emission measurements | ±30 min from measurement time | 10 days (CH$_3$I), 3 months (CHBr$_3$), 1.5 years (CH$_2$Br$_2$) | 10,000 per measurem. |
| *Indian Ocean* | Forward; VSLS tracers | 1°x1° grid at sea surface; 50°E - 80°E, 20°S - 10°N | Every day from July 1-31, 2000-2015 | 3 months | 29,791 x 16 years |

**Table 2: CHBr$_3$, CH$_2$Br$_2$, and CH$_3$I water concentrations, air mixing ratios, and calculated emissions for the OASIS Indian Ocean cruise. The table lists the average value of all measurements and 1 standard deviation. The brackets give the range of measurements.**

| VSLS | Water concentration [pmol L$^{-1}$] | Air mixing ratio [ppt] | Emission [pmol m$^{-2}$ hr$^{-1}$] |
|---|---|---|---|
| CHBr$_3$ | 8.4 ± 14.2 [1.3 – 33.4] | 1.20 ± 0.35 [0.68 – 2.97] | 910 ± 1160 [-100 – 9,630] |
| CH$_2$Br$_2$ | 6.7 ± 12.6 [0.6 – 114.3] | 0.91 ± 0.08 [0.77 – 1.20] | 930 ± 2000 [-70 – 19,960] |
| CH$_3$I | 3.4 ± 3.1 [0.2 – 16.4] | 0.84 ± 0.12 [0.57 – 1.22] | 460 ± 430 [5 – 2,090] |





**Table 3: CHBr₃, CH₂Br₂, and CH₃I mean emissions [pmol m⁻² hr⁻¹] for several cruises and observational and model based climatological studies. Abbreviations: IO = Indian Ocean, OLS = Ordinary Least Square Method**


| Study | Cruise / Region | CHBr$_3$ | CH$_2$Br$_2$ | CH$_3$I |
|---|---|---|---|---|
| This study | OASIS / West IO | 910 | 930 | 460 |
| Chuck et al., 2005 | ANT XVIII/1 / Tropical Atlantic | 125 | | 625 |
| Tegtmeier et al., 2012, 2013 | TransBrom / West Pacific Ocean | 608 | 164 | 320 |
| Hepach et al., 2014 | DRIVE / Tropical Atlantic | 787 | 341 | 254 |
| Hepach et al., 2015 | MSM 18/3 / Equatorial Atlantic | 644 | 187 | 425 |
| Hepach et al., 2016 | M91 / Peruvian Upwelling | 130 | 273 | 954 |
| Fuhlbruegge et al., 2016b | SHIVA / South China Sea | 1486 | 405 | 433 |
| Quack and Wallace, 2003 | Global open ocean | 625 | | |
| Yokouchi et al., 2005 | Global open ocean | | 119 | |
| Butler et al., 2007 | Tropical ocean | 379 | 108 | 541 |
| Carpenter et al. 2009 | Atlantic open ocean | 367 | 158 | |
| Bell et al., 2002 | Global ocean | | | 670 |
| Warwick et al., 2006 | Tropics, Scenario 5 | 580 | | |
| Liang et al., 2010 | Tropics, open ocean, Scenario A | 854 | 81 | |
| Ordoñez et al., 2012 | Tropics | 956 | | |
| Ziska et al., 2013 | IO equator, OLS | ≈500 | ≈500 | ≈250 |
| Ziska et al., 2013 | IO subtropics, OLS | ≈250 | ≈250 | ≈500 |
| Stemmler et al., 2013 | Tropical Atlantic Ocean | | | ≈500 |
| Stemmler et al., 2014 | IO equator | ≈200 | | |



**Table 4: Mean Flexpart emission, entrainment at 17 km, transport efficiency, and transit half-life for CHBr$_3$, CH$_2$Br$_2$ and CH$_3$I for the mean and different transport regimes of the OASIS cruise.**

| VSLS | Transport regime | Flexpart emission [µmol] | Emissions by regime [%] | Transport efficiency [%] | Flexpart entrainment [nmol] | Entrainment by regime [%] | Transit half-life [days] |
|---|---|---|---|---|---|---|---|
| CHBr$_3$ | Cruise mean | **0.43** | - | **1.38** | **5.5** | - | **21** |
| | Westerlies | 0.49 | 32 | 0.36 | 1.83 | 9 | 32 |
| | Transition | 0.36 | 24 | 0.58 | 2.05 | 11 | 24 |
| | Monsoon Circulation | 0.51 | 34 | 2.08 | 10.70 | 57 | 15 |
| | Local Convection | 0.15 | 10 | 2.86 | 4.31 | 23 | 10 |
| CH$_2$Br$_2$ | Cruise mean | **0.43** | - | **6.38** | **23.6** | - | **86** |
| | Westerlies | 0.71 | 48 | 3.99 | 28.8 | 35 | 112 |
| | Transition | 0.32 | 22 | 4.89 | 15.0 | 19 | 114 |
| | Monsoon Circulation | 0.31 | 21 | 8.22 | 26.2 | 31 | 63 |
| | Local Convection | 0.14 | 9 | 8.83 | 12.7 | 15 | 57 |
| CH$_3$I | Cruise mean | **0.22** | - | **0.25** | **0.7** | - | **6** |
| | Westerlies | 0.15 | 18 | 0.00 | 0.00 | 0 | 9 |
| | Transition | 0.11 | 13 | 0.00 | 0.00 | 0 | 9 |
| | Monsoon Circulation | 0.28 | 33 | 0.28 | 0.74 | 21 | 7 |
| | Local Convection | 0.31 | 36 | 0.95 | 2.77 | 79 | 1 |




**Table 5: CHBr$_3$ entrainment at 17 km for different ocean regions using the same transfer coefficient for the emission calculations and Flexpart model set-up (Sect. 2.2). The table lists the average value and 1 standard deviation. The brackets give the range of single calculations.**

| Ocean region | Campaign information | Flexpart emission [nmol] | Flexpart entrainment [nmol] | Transport efficiency [%] |
|---|---|---|---|---|
| **West Indian Ocean** | OASIS, July 2014 (this study) | 430 ± 520 [4 – 4130] | 5.5 ± 7.5 [0.0 – 50.1] | 1.4 ± 1.0 [0.1 – 3.9] |
| **Open West Pacific** | TransBrom, Oct. 2009 (Krüger and Quack, 2013) | 190 ± 300 [0 – 5680] | 7.1 ± 10.4 [0.0 – 61.8] | 4.4 ± 1.6 [1.9 – 8.8] |
| **Coastal West Pacific** | SHIVA, Nov. 2011 (Fuhlbrügge et al., 2016b) | 610 ± 720 [1 – 5680] | 48.4 ± 52.1 [0.7 – 250.1] | 7.9 ± 3.7 [3.2 – 20.2] |
| **Equatorial Atlantic** | MSM18/3, June 2011 (Hepach et al., 2015) | 320 ± 400 [2 – 1910] | 2.7 ± 3.2 [0.0 – 14.2] | 0.9 ± 0.2 [0.5 – 1.4] |

**Table 6: Entrainment of CHBr$_3$, CH$_2$Br$_2$ and CH$_3$I tracer at 17 km altitude through different transport regimes from the West Indian Ocean release box. (Note, transit half-lifes differ from Table 4 because of the different model setups.)**

| Tracer | Transport regime | Mean transport efficiency [%] | Transit half-life [days] | Interannual correlation with *Total* entrainment | Interannual correlation with *Local Convection* entrainment |
|---|---|---|---|---|---|
| **CHBr$_3$** | *Total* | 1.86 | 8.5 | 1.00 | 0.39 |
| | *Local Convection* | 0.20 | 2.5 | 0.39 | 1.00 |
| | *Monsoon Circulation* | 0.50 | 6.0 | 0.54 | -0.23 |
| **CH$_2$Br$_2$** | *Total* | 5.88 | 27.2 | 1.00 | 0.17 |
| | *Local Convection* | 0.28 | 3.8 | 0.17 | 1.00 |
| | *Monsoon Circulation* | 1.11 | 13.3 | 0.56 | -0.11 |
| **CH$_3$I** | *Total* | 0.42 | 1.9 | 1.00 | 0.87 |
| | *Local Convection* | 0.14 | 1.6 | 0.87 | 1.00 |
| | *Monsoon Circulation* | 0.09 | 1.0 | -0.06 | -0.29 |



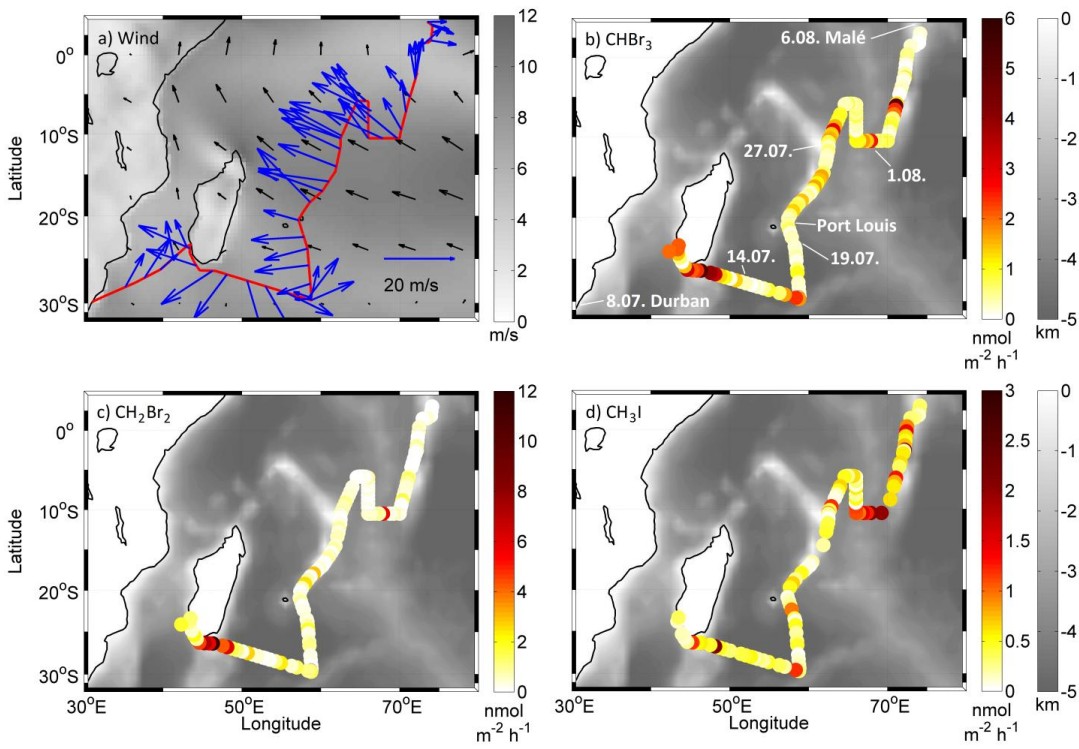

Figure 1: a) July 2014 average wind speed (grey shading) and direction (black) from ERA-Interim and 10 min mean wind speed (blue arrows) from ship sensors; b) CHBr$_3$, c) CH$_2$Br$_2$, and d) CH$_3$I emissions derived from OASIS cruise July-August 2014 and bathymetry.







Figure 2: a) Surface air temperature (SAT), sea surface temperature (SST), b) wind speed and direction measured by ship sensors during the OASIS cruise in the Indian Ocean. c) Water concentration, d) atmospheric mixing ratio and e) emission of CHBr₃, CH₂Br₂, and CH₃I. The grey line denotes the harbor stop at Port Louis, Mauritius, July 20-23, 2014. Also note the nonlinear left y-axes in c) and e).



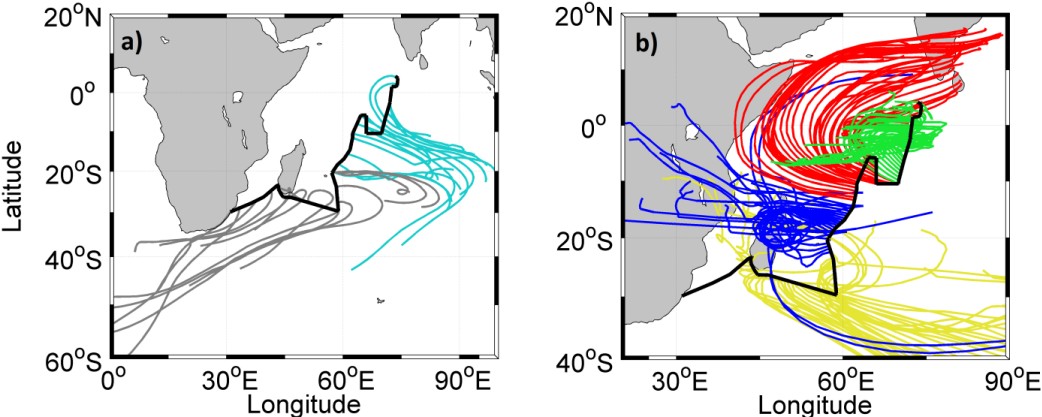


**Figure 3: a) Flexpart five day backward trajectories for *OASIS backward* setup, averaged for n=50 trajectories, starting from ship positions daily at 12 UTC between July 8 and August 7, 2014. The Southern Ocean (grey) and the Open Indian Ocean (turquoise) are source regions for air measured during the cruise. b) Flexpart ten day forward trajectories for *OASIS* setup, averaged for n=1000**

**trajectories, starting at the ship positions of simultaneous VSLS measurements. Trajectories are colored according to their transport regimes: Westerlies, Transition, Monsoon Circulation, and Local Convection.**





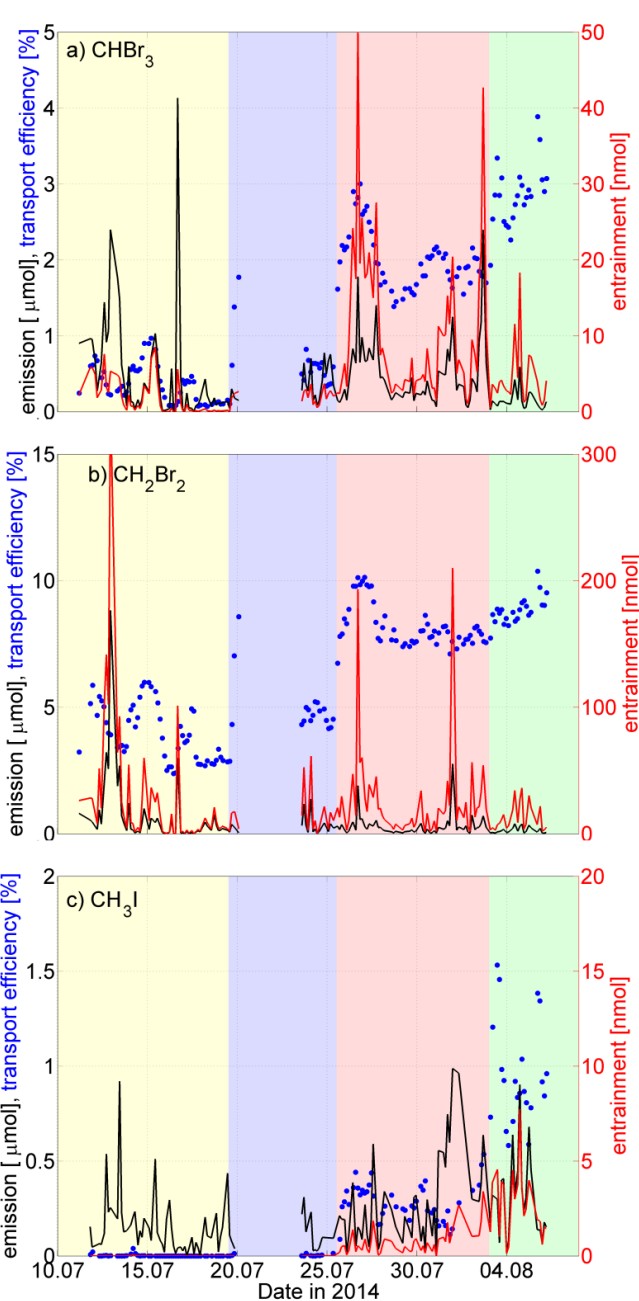

**Figure 4: CHBr₃, CH₂Br₂, and CH₃I emission, entrainment at 17 km and transport efficiency for measurements from the OASIS cruise. The background shading highlights the transport regimes: Westerlies, Transition, Monsoon Circulation, and Local Convection.**



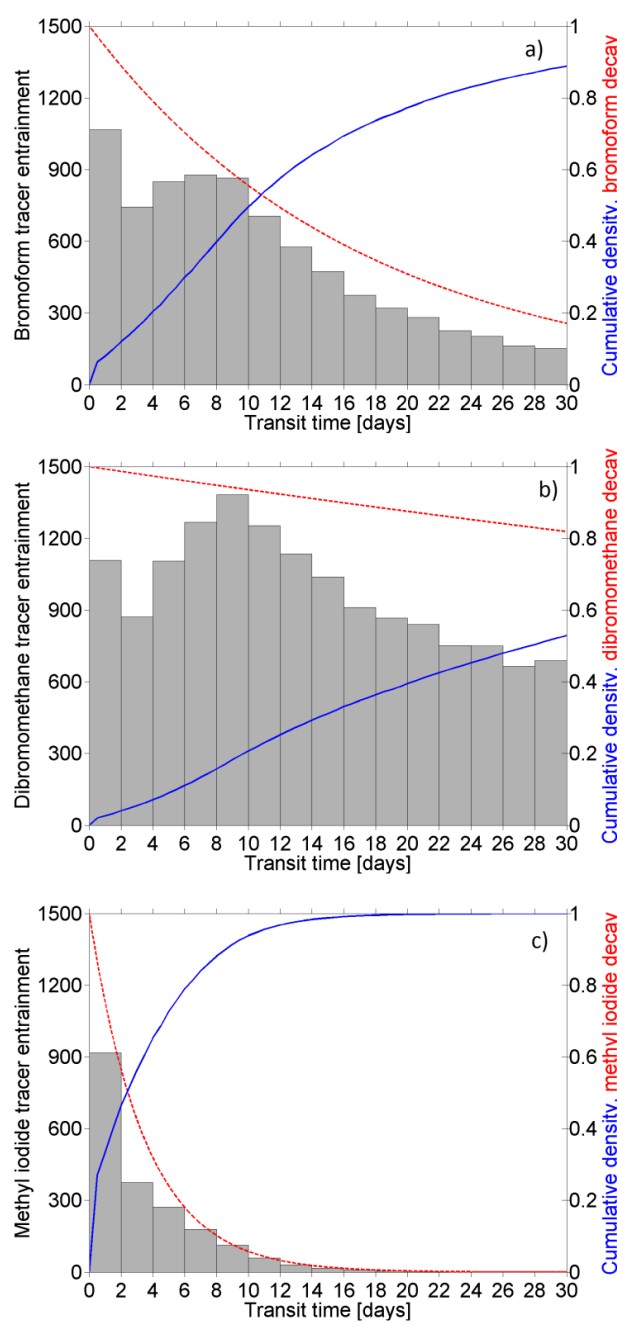

**Figure 5: VSLS transit time distribution for entrainment at 17 km of a) CHBr₃, b) CH₂Br₂, and c) CH₃I tracers released in July 2000-2015. Entrained tracer per time interval of 2 days is given as number (grey bars). The blue line gives the cumulative distribution and denotes the transit half-life. The red line shows the decay of the tracers during the transport simulation.**



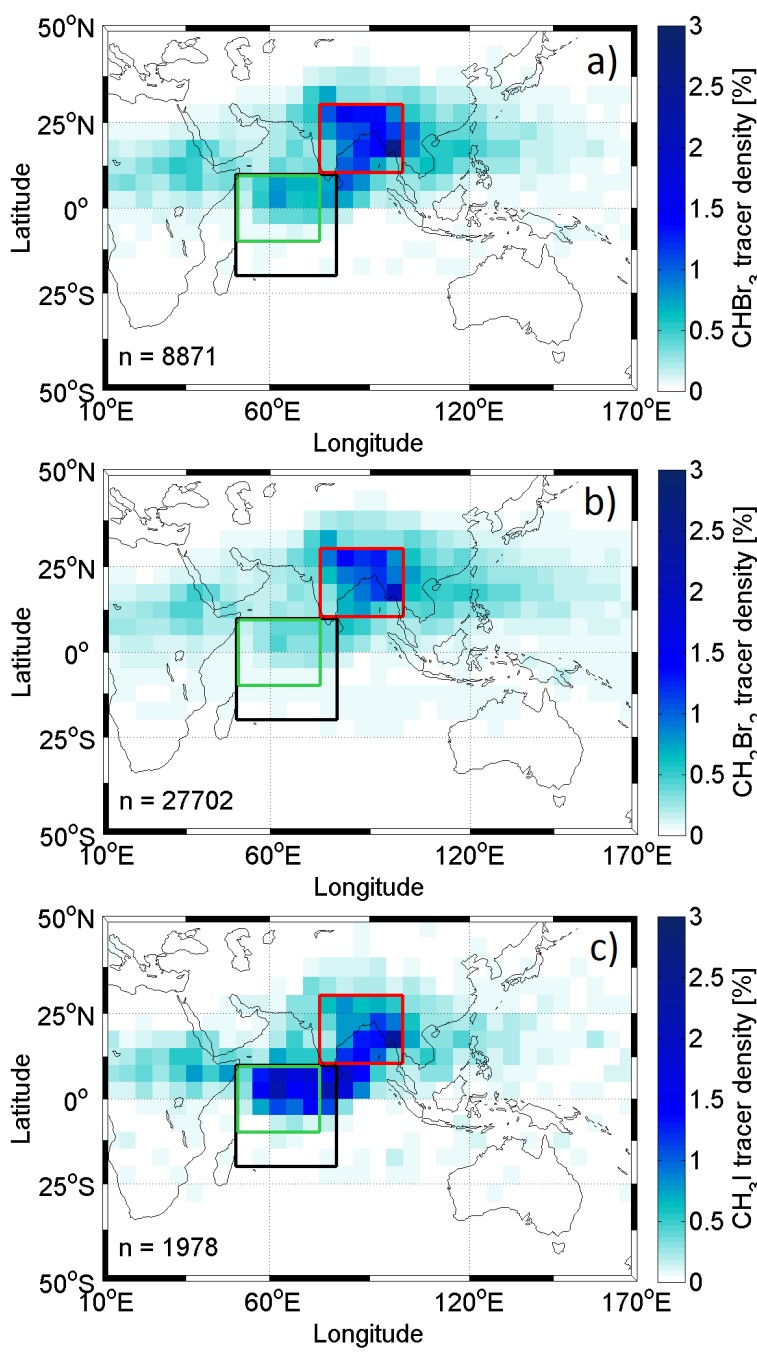

**Figure 6: Density at 17 km of a) CHBr₃, b) CH₂Br₂, and c) CH₃I tracer on a 5°x5° grid that is released from the West Indian Ocean surface (black box) in July, 2000-2015. Colored boxes show the entrainment regions of the Local Convection and Monsoon Circulation regimes.**





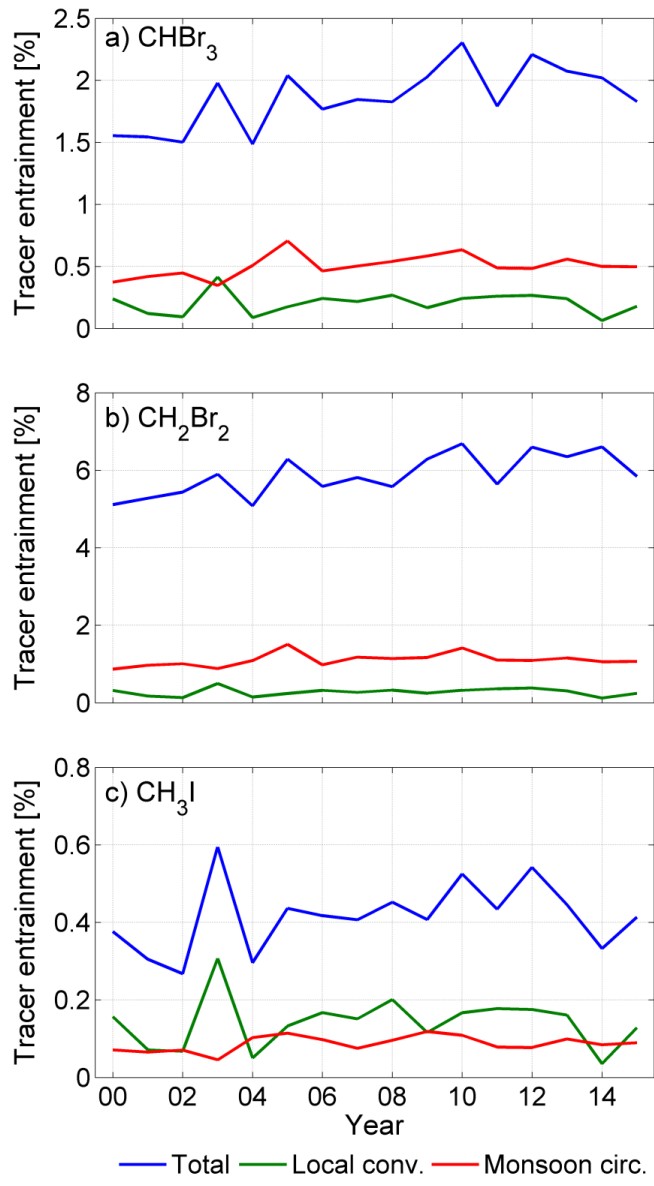

**Figure 7: a) CHBr₃, b) CH₂Br₂, and c) CH₃I tracer entrainment at 17 km from trajectories released from the West Indian Ocean surface box in July 2000 - 2015. The entrainment is evaluated for three regions: Total, Local Convection, and Monsoon Circulation (see Fig. 6).**

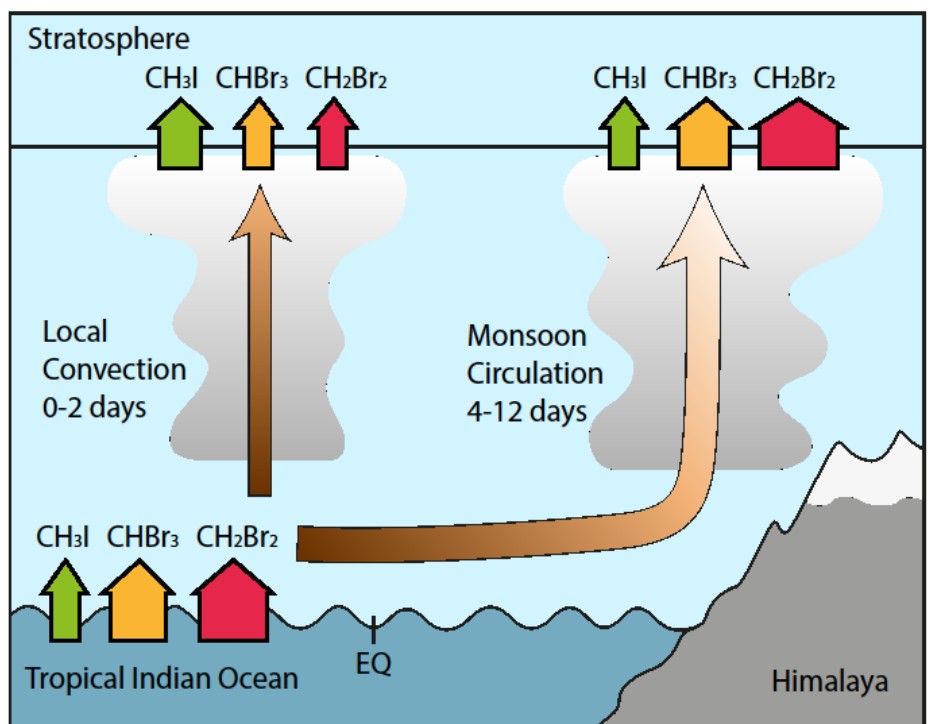


**Figure 8: Schematic illustration of emission, transport pathways and timescales, and entrainment of CH$_3$I, CHBr$_3$, and CH$_2$Br$_2$ tracer from the tropical West Indian Ocean to the stratosphere during the Asian summer monsoon.**