# Peer review of "Delivery of halogenated very short-lived substances from the West Indian Ocean to the stratosphere during Asian summer monsoon"

_Atmospheric Chemistry and Physics, 2017_

## Referee Comment (RC1) · Anonymous Referee #1 · 10 Feb 2017

Review for "Delivery of halogenated very short-lived substances from the West Indian Ocean to the stratosphere during the Asian summer monsoon" Alina Fiehn, et al. ACP-2017-8

Summary

The manuscript by Fiehn et al. presents observations of very short-lived species (VSLS) obtained from ship-based measurements during a cruise on the research vessel Sonne in the subtropical and tropical Indian Ocean in July and August 2014. The species observed are $CH_3I$, $CHBr_3$ and $CH_2Br_2$, all of which have potentially a strong impact on stratospheric chemistry and climate. Measurements of VSLS are sparse and show large variability. Attempts to create global emission estimates by creating

observation based climatologies or modeling are characterized by large uncertainties.

The observations presented consist of air and water samples and were obtained together with data from radio sondes launched during the cruise. Utilizing these local meteorological data compound specific transfer coefficients and subsequently air-sea fluxes of VSLS were calculated.

The paper does not only present observations, but includes transport simulations utilizing trajectories computed with the Lagrangian model Flexpart. The aim of this part of the study is on the identification of the source regions of the air masses observed as well as transport pathways of VSLS from the boundary layer into the stratosphere, followed by an estimation of the amount of the above mentioned species entrained into the stratosphere during the Asian summer monsoon. Different model set ups are used covering backward, forward and domain filling forward scenarios. In addition the study is applied to the results obtained during other measurement campaigns and extended to 16 years to specify inter-annual variability.

The results and conclusions presented show two major pathways of transport of VSLS from the West Indian Ocean boundary layer into the stratosphere, namely local convection on a time scale of 0-2 days and upward transport inside the Asian monsoon circulation over Nortehern India and the Bay of Bengal on a time scale of 6 to 13 days.

General

In general the manuscript is an organized and methodical sound paper. Starting from an introduction that embeds the studied processes into the current scientific background regarding the importance of biogenic VSLS for the stratospheric bromine and iodine budget as well as their role in stratospheric chemistry, and after having presented their observations, the authors focus on the role of vertical transport for the entrainment of VSLS into the stratosphere, the related problem of transit times and life times for different compounds and major pathways.

Aiming at these issues, the authors utilize the Flexpart trajectory model, a state-of-the-art transport model often used for investigations of transport processes in the troposphere. The model was driven by ERA Interim meteorological data with a 1 x 1 degree horizontal resolution. In first simple setups (OASIS backward and forward), the origin of the air masses probed along the ship track and their future trajectories are investigated. The OASIS forward trajectories already indicate the existence of different transport regimes, with upward transport within local convection over the West Indian Ocean and inside the Asian monsoon circulation above Northern India and the Bay of Bengal being the most prominent features. Entrainment into the stratosphere is supposed at 17 km height.

This leads to my first critical remark: The authors claim, that the value of 17 km height for the location of the tropopause represents the average cold point tropopause during the cruise and for the whole tropical region. This may well be true, but as one can depict from the figure S2, there is quite strong variability around the mean value, especially the more to the north one gets. Furthermore the authors show with quite a lot of effort results for different other heights to substantiate the choose of this value (section 5 and figure S3). But why not using a tropopause location computed from the ERA Interim data, the same data underlying the transport calculation? I am not doubting the general location of the maximum entrainment areas, but using the ERA Interim based tropopause would be much more convincing and more consistent.

Speaking of this, ERA Interim provides much higher horizontal resolution than used. Why not using it for better, more accurate trajectory results, better resolved convection and better tropopause location? Even the Flexpart parametrizations for vertical transport and convection could benefit from this.

To investigate more on stratospheric entrainment and the role of the West Indian Ocean a third forward trajectory model run is started with domain filling trajectories from a rectangular area over the area of interest (Indian Ocean setup). In this context now VSLS tracers are used, which undergo an exponential decay according to certain tropical

tropospheric life times. This is in contrast to the vertical life time profile used for the simple OASIS setups, and I am not sure, why there are two different life times used. May be the authors can comment on this.

Nevertheless the results from this run and the results gained for different other campaigns and regional areas are quite interesting and plausible. The analysis of transit times, transport efficiencies and transit half times provides a good insight into the interplay of rapid local convection and slower upward motion (as inside monsoon circulations) and the impact on species with life times being much smaller than or comparable to the transport time.

There is one more remark about the investigtions with respect to the spatial varability of the stratospheric entrainment: In line 502 the definition of two core entrainment areas are mentioned, which are assumed to be evenly sized (and are shown in Fig. 6). It should be noted that these two regions may be evenly sized in grid space (that is 20 x 25 degrees), but not in area. Furthermore just by looking at the plots, one could think of moving the core entrainment box for the local convection 5 degrees more to the east for a better capture of entrainment. But maybe these boxes are chosen to be similar to Chen et al.(2012). If this is the case, it should be mentioned.

In the last part of the study the simulation with the Indian Ocean setup is extended to 16 years to specify inter-annual variability. To quantify the influence of different transport regimes (local convection and Asian monsoon) on the total entrainment, the respective time series are correlated by using Pearsons correlation coefficient. It should be noted, that for all values not -1 or 1 Pearsons r ist not meaningful, as long as there is no linearity between the two times series and/or if the values are not normally distributed. If there is a linear relationship, than correlation coefficients of 0.54 and 0.56 only explain 29% and 32% of the observed variance, meaning that roughly 70% of the variance is not explained. Even for a value of 0.87 (as for CH3I) just 75% of the variance are explained. To decide, whether these values are significant, you need to do a t test. A much more robust method, which does not imply linearity, but only monotonicity,

is Spearmans rank correlation coefficient. There is a similar method introduced by Kendall. I would recommend to use one of these rank correlation coefficients, which is fairly easy to do, but would give more meaningful results.

Within the concluding section the main results are summarized clearly, followed by implications of the papers findings for global emission estimates for short-lived species and the influence of different transport pathways on the stratospheric entrainment of short-lived species. The authors discuss some potential problems regarding the calculation of vertical transport and uncertainties in the determination of the stratospheric entrainment with respect to the average cold point tropopause. Again, I would like to stress the point of calculating the local tropopause height directly from the used ERA Interim data, preferably from data with higher horizontal resolution.

Summarizing, the paper is well-written and presents an important contribution to our understanding of transport of the short-lived species from the boundary layer over the West Indian Ocean into the stratosphere. It should be published after some minor revisions.

Additional comments

The values shown in figure 6 are labeled as tracer density (given in percent). Is this meant to be the same as tracer entrainment?
* * *

---

## Referee Comment (RC2) · Anonymous Referee #2 · 21 Mar 2017

This paper presents observations of atmospheric and oceanic CHBr3, CH2Br2 and CH3I from a cruise in the tropical West Indian Ocean. A trajectory model is used to estimate how much of the derived oceanic emissions of these short-lived compounds reach the stratosphere. Comparisons are made with observations from other cruises and the impact of interannual variations in meteorology is also assessed.

Overall I think this is a good paper and should be published in ACP. There is considerable interest in short-lived halocarbon emissions and the cruise presented here provides important additional information. The stratospheric input is quantified and presented with a range of model-based metrics which allow comparison with other studies. The uncertainties/limitations of the modelling is described.

[Figure]

I have only minor comments:

Line 130. The word 'project' is not correct here. (Also line 466).

Line 136. '..during the Asian summer monsoon season'? (Missing words?)

Line 193. Do you mean 154 samples every 3 hours, or 154 overall which are spaced about every 3 hours?

Line 257. 'July 2000-2015'. Should be rewritten to clarify it is for July during those years.

Line 283. Can you show these differences with respect to the ECWMF winds somehow? Can you add the time varying ECMWF winds in Figure 2, if there is a discrepancy to discuss?

Line 291. Give the lifetime of butane so the reader can judge what this is testing.

Line 301. Change 'lower' to 'smaller'. (There are many other places where I would suggest changing higher to larger, when higher can be confused with meaning higher altitude).

Section 3.3. This section compares the emission values between the cruises, but I think it is missing an overall synthesis or discussion about what these differences tell us about the different regimes or techniques. I would suggest adding a paragraph after line 389.

Line 398. It would be interesting to see some plot of the vertical distribution of tracers in the different transport regimes.

Line 413. I don't understand the definition of transit half-life. It might be the use of 'has reached 17km.' All entrained tracer reaches 17km? It seems like Figure 5 would help but that comes later. For Figure 5 can you add a symbol on the blue line at the half-life value? (If I have understood correctly).

Line 416. Table 4. Please check all the values in Table 4. I tried to check my understanding of Transport Efficiency by dividing entrainment by emission. E.g. for CHBr3 cruise mean: 5.5/430 = 1.28%. Not 1.38. I tried other values and there seemed to be differences (23.6/430 = 5.49% and not 6.38%). What is wrong? Also, it would help if the text used the same precision as the table (e.g. line 419 say 1.38% and not 1.3%, or is it 1.28%?).

Figure 5. What are the units of the left-hand y axis?

[Figure]

---

## Author Comment (AC2) · 19 Apr 2017

We would like to thank Reviewer 2 for the suggestions to further improve the manuscript. Below you find our answers to your specific points. The reviewer's comment is marked with 'RC:' and is written in quotes, our answer in normal font.

RC: "Overall I think this is a good paper and should be published in ACP. There is considerable interest in short-lived halocarbon emissions and the cruise presented here provides important additional information. The stratospheric input is quantified and presented with a range of model-based metrics which allow comparison with other studies. The uncertainties/limitations of the modeling are described."

We thank the reviewer for this very positive review.

Answers to the specific comments of the reviewer:

RC: “Line 130: The word ‘project’ is not correct here. (Also line 466).”

In line 130, we changed ‘project’ to ‘model’. In line 466, ‘projected’ was changed to ‘determined’.

RC: “Line 136: ‘..during the Asian summer monsoon season’? (Missing words?)”

The term summer monsoon already includes one season. We thus think that the word “season” is not necessary in the manuscript.

RC: “Line 193: Do you mean 154 samples every 3 hours, or 154 overall which are spaced about every 3 hours?”

We mean 154 overall samples spaced about every 3 hours. We added “overall” and “spaced about” to the sentence.

RC: “Line 257: ‘July 2000-2015’. Should be rewritten to clarify it is for July during those years.”

Done.

RC: “Line 283: Can you show these differences with respect to the ECWMF winds somehow?”

Figure 1a shows the ECMWF monthly mean winds as black arrows and the in situ ship measurements as blue arrows. Thus, we think differences are directly visible in the figure. See also our answer to your next comment.

RC: “Can you add the time varying ECMWF winds in Figure 2, if there is a discrepancy to discuss?”

The comparison of time varying ERA-Interim winds and in situ ship winds can be found in Fig. S1 in the supporting material. In line 285, we added the sentence: “Surface

winds from in situ ship measurements, radiosondes, and time varying ERA-Interim data show good agreement (Fig. S1)."

RC: "Line 291: Give the lifetime of butane so the reader can judge what this is testing."

The atmospheric lifetime of butane is estimated to 2.5 days considering the reaction with OH and ozone (Finlayson-Pitts and Pitts, 2000). We added to line 291: "(lifetime 2.5 days (Finlayson-Pitts and Pitts, 2000))".

RC: "Line 301: Change 'lower' to 'smaller'. (There are many other places where I would suggest changing higher to larger, when higher can be confused with meaning higher altitude)."

Yes, we agree with the reviewer and changed the words "lower" and "higher" in several places in Sect. 3.2 and 3.3 to "smaller" and "larger", respectively.

RC: "Section 3.3.: This section compares the emission values between the cruises, but I think it is missing an overall synthesis or discussion about what these differences tell us about the different regimes or techniques. I would suggest adding a paragraph after line 389."

We added the following paragraph at line 389: "In general, the emissions measured during the OASIS cruise in the subtropical and tropical West Indian Ocean were as large or larger than in other tropical open ocean cruises or studies. Especially CH2Br2 emissions during the OASIS cruise were larger than any previous emission estimates. The West Indian Ocean seems to be a region with significant contribution to the global open ocean VSLS emissions, especially in boreal summer when wind speeds are high because of the southwest monsoon circulation."

RC: "Line 398: It would be interesting to see some plot of the vertical distribution of tracers in the different transport regimes."

We show the vertical distribution of bromoform from the OASIS cruise for the four transport regimes averaged over the time of the model calculation in the below Fig. R1.

These profiles confirm that the bromoform emissions in the Westerlies and Transitions regime stay mostly below 5 km height, while in the Monsoon Circulation and Local Convection regimes bromoform is more uniformly distributed in the troposphere. We added this figure to the supporting material and explanatory text in line 404: “The different uplift heights are reflected in the vertical distribution of bromoform in each transport regime (Fig. S2).”

RC: “Line 413: I don’t understand the definition of transit half-life. It might be the use of ‘has reached 17km.’ All entrained tracer reaches 17km? It seems like Figure 5 would help but that comes later. For Figure 5 can you add a symbol on the blue line at the half-life value? (If I have understood correctly).”

We used the term “entrained tracer” as a synonym for “has reached 17 km”. We changed the explanation to: “. . .transit half-life, which is the time after which half of the total amount of entrained tracers have been entrained above 17 km altitude.” We also added a half-life marker in Fig. 5.

RC: “Line 416: Table 4: Please check all the values in Table 4. I tried to check my understanding of Transport Efficiency by dividing entrainment by emission. E.g. for $CHBr_3$ cruise mean: 5.5/430 = 1.28%. Not 1.38. I tried other values and there seemed to be differences (23.6/430 = 5.49% and not 6.38%). What is wrong? Also, it would help if the text used the same precision as the table (e.g. line 419 say 1.38% and not 1.3%, or is it 1.28%?).”

Thanks for checking Table 4 carefully! Your understanding of the transport efficiency is correct. There were slips in Table 4, because of a former version of the manuscript. We recalculated transport efficiencies, corrected the tables and adjusted the precision in the table and Sect. 3.4.

RC: “Figure 5: What are the units of the left-hand y axis?”

The unit on this axis is number. We added the unit to the y-axis description of Fig. 5.

References:

Finlayson-Pitts, B., and Pitts, J.: Chemistry of the upper and lower atmosphere: Theory, experiments and applications, Academic, US, 2000.
* * *
[Figure]

**Fig. 1.** Time-averaged vertical distribution of bromoform in the four transport regimes (Figure 3b).

---

## Author Response (AR1)

**Author's Response**

**Delivery of halogenated very short-lived substances from the West Indian Ocean to the stratosphere during Asian summer monsoon**

Alina Fiehn[1, 2], Birgit Quack[2], Helmke Hepach[2,*], Steffen Fuhlbrügge[2], Susann Tegtmeier[2], Matthew Toohey[2], Elliot Atlas[3], Kirstin Krüger[1]

[1]*Meteorology and Oceanography Section, Department of Geosciences, University of Oslo, Oslo, Norway*
[2] *GEOMAR Helmholtz Centre for Ocean Research Kiel, Kiel, Germany*
[3] *Rosenstiel School of Marine and Atmospheric Science, University of Miami, Miami, USA*
* *now at: Environment Department, University of York, York, United Kingdom*

**Answers to Reviewer Comment 1**

We would like to thank Reviewer 1 for the comments to improve the manuscript. Below, you find our answers to the specific points. The reviewer's comment is written in *italic letters*, our answer in normal font.

*Summarizing, the paper is well-written and presents an important contribution to our understanding of transport of the short-lived species from the boundary layer over the West Indian Ocean into the stratosphere. It should be published after some minor revisions.*

We thank the reviewer for this very positive review.

Answers to the specific comments of the reviewer:

*This leads to my first critical remark: The authors claim, that the value of 17 km height for the location of the tropopause represents the average cold point tropopause during the cruise and for the whole tropical region. […] But why not using a tropopause location computed from the ERA Interim data, the same data underlying the transport calculation? I am not doubting the general location of the maximum entrainment areas, but using the ERA Interim based tropopause would be much more convincing and more consistent.*

We tested several definitions for stratospheric entrainment for our trajectory runs. Flexpart includes the lapse rate tropopause (LRT) calculation after the WMO (1957) definition. Using our Flexpart/ERA-Interim trajectory runs this tropopause lies lower (14-15 km, see Fig. R1 below) than the one we observed during the cruise (17.0 km Fig S2 from paper) and the one inferred from satellite measurements during JJA at the equator (LRT 16.2 km, cold point tropopause (CPT) 16.7 km (Munchak and Pan, 2014)). In the Asian monsoon anticyclone the tropopause lies even higher between 17.2 (LRT) and 17.6 km (CPT) height for a 4 year mean (Munchak and Pan, 2014). A temperature threshold of 192 K (average CPT temperature over the Indian monsoon region in JJA (Kim and Son, 2012)) lead to a tropopause height around 16.5 km (Fig. R1). Thus, we decided to use 17 km height as a conservative approximation to the tropical tropopause height over the Indian Ocean and Asia during boreal summer and added other entrainment altitudes (13, 15, 18 km) in the supplementary material.

[Figure]

**Figure R1:** Height of trajectories at entrainment locations using different tropopause definitions. Dashed lines show the lapse rate tropopause from ERA-Interim, and solid lines the height of trajectories reaching temperatures below 192 K. 30SN means stratospheric entrainment between 30°S and 30°N and so on.

*Speaking of this, ERA Interim provides much higher horizontal resolution than used. Why not using it for better, more accurate trajectory results, better resolved convection and better tropopause location? Even the Flexpart parameterizations for vertical transport and convection could benefit from this.*

The spatial resolution of the current available **ECMWF reanalysis from 1979 onwards**, ERA-Interim, is approximately 80 km in grid space (T255 triangular truncation) on 60 vertical levels from the surface up to 0.1 hPa. Thus the chosen 1˚x1˚ grid resolution (111 km at the equator) is very close to the original model resolution. Higher spatial and vertical resolution ECMWF data is only available from the operational model with T1279 and L137 (since 2010 and 2013 respectively). However, the high resolution operational model was not suitable for our study on a longer interannual time series due to regular changes of the operational ECWMF weather model in order to improve the forecast.

*To investigate more on stratospheric entrainment and the role of the West Indian Ocean a third forward trajectory model run is started with domain filling trajectories from a rectangular area over the area of interest (Indian Ocean setup). In this context now VSLS tracers are used, which undergo an exponential decay according to certain tropical tropospheric life times. This is in contrast to the vertical life time profile used for the simple OASIS setups, and I am not sure, why there are two different life times used. May be the authors can comment on this.*

The two calculation methods differ slightly from each other in their timing of the calculation of compound decay. For the first method, the VSLS **emissions** are attached to the trajectories and the decay is calculated online with the model. The height of the particle position is known and can be used for determination of lifetime according to the vertical profile. For the second approach, the VSLS **tracer** transport is attributed to a trajectory after the model has been run by considering the transport time. We added the following explanatory sentence in line 264: "The use of VSLS tracers allows us to evaluate one model run for different compounds with varying lifetimes."

*There is one more remark about the investigations with respect to the spatial variability of the stratospheric entrainment: In line 502 the definition of two core entrainment areas are mentioned, which are assumed to be evenly sized (and are shown in Fig. 6). It should be noted that these two regions may be evenly sized in grid space (that is 20 x 25 degrees), but not in area. Furthermore just by looking at the plots, one could think of moving the core entrainment box for the local convection 5 degrees more to the east for a better capture of entrainment. But maybe these boxes are chosen to be similar to Chen et al. (2012). If this is the case, it should be mentioned.*

Thanks for this comments. Yes it is true. Geographically, the entrainment areas are not evenly sized. We reformulated to "… and to be evenly sized in grid space" in line 502. The entrainment areas were not chosen to be similar to Chen et al. (2012). We will furthermore move the Local Convection entrainment area 5˚ to the east. This will also lead to changes in Fig. 6, Table 6 and the numbers in Sect. 4.2.

*In the last part of the study the simulation with the Indian Ocean setup is extended to 16 years to specify inter-annual variability. To quantify the influence of different transport regimes (local convection and Asian monsoon) on the total entrainment, the respective time series are correlated by using Pearsons correlation coefficient. It should be noted, that for all values not -1 or 1 Pearsons r is not meaningful, as long as there is no linearity between the two times series and/or if the values are not normally distributed. If there is a linear relationship, than correlation coefficients of 0.54 and 0.56 only explain 29% and 32% of the observed variance, meaning that roughly 70% of the variance is not explained. Even for a value of 0.87 (as for CH$_3$I) just 75% of the variance are explained. To decide, whether these values are significant, you need to do a t test. A much more robust method, which does not imply linearity, but only monotonicity is Spearman's rank correlation coefficient. There is a similar method introduced by Kendall. I would*

*recommend to use one of these rank correlation coefficients, which is fairly easy to do, but would give more meaningful results.*

Thanks for this discussion. We checked all our time series for normal distribution. They are normally distributed and thus Pearson correlations can be applied. Then we calculated p-values to infer the significance of the correlations in Table 6. We marked all 95% significant correlations in bold face. We also calculated the Spearman's rank correlation coefficients. These differed less than 0.1 from the Pearson correlation coefficients and thus we would like to continue using the Pearson correlation. The following sentence is added to Sect. 2.2:" We calculated the p-value to determine the 95% significance level of the correlations."

*The values shown in figure 6 are labeled as tracer density (given in percent). Is this meant to be the same as tracer entrainment?*

Yes, it is meant to be tracer entrainment, but the relative distribution of this tracer entrainment. So it could also be named relative tracer entrainment, or entrainment density. We changed the label in Fig. 6 to "entrainment density".

**Citations**

Kim, J., and Son, S.-W.: Tropical Cold-Point Tropopause: Climatology, Seasonal Cycle, and Intraseasonal Variability Derived from COSMIC GPS Radio Occultation Measurements, Journal of Climate, 25, 5343-5360, 10.1175/jcli-d-11-00554.1, 2012.
Munchak, L. A., and Pan, L. L.: Separation of the lapse rate and the cold point tropopauses in the tropics and the resulting impact on cloud top-tropopause relationships, Journal of Geophysical Research: Atmospheres, 119, 7963-7978, 10.1002/2013jd021189, 2014.
WMO: Definition of the tropopause, WMO Bull., 6, 136, 1957.

**Answers to Reviewer Comment 2**

We would like to thank Reviewer 2 for the suggestions to further improve the manuscript. Below you find our answers to your specific points. The reviewer's comment is written in *italics*, our answer in normal font.

*Overall I think this is a good paper and should be published in ACP. There is considerable interest in short-lived halocarbon emissions and the cruise presented here provides important additional information. The stratospheric input is quantified and presented with a range of model-based metrics which allow comparison with other studies. The uncertainties/limitations of the modeling are described.*

We thank the reviewer for this very positive review.

Answers to the specific comments of the reviewer:

*Line 130: The word 'project' is not correct here. (Also line 466).*

In line 130, we changed 'project' to 'model'. In line 466, 'projected' was changed to 'determined'.

*Line 136: '..during the Asian summer monsoon season'? (Missing words?)*

The term summer monsoon already includes one season. We thus think that the word "season" is not necessary in the manuscript.

*Line 193: Do you mean 154 samples every 3 hours, or 154 overall which are spaced about every 3 hours?*

We mean 154 overall samples spaced about every 3 hours. We added "overall" and "spaced about" to the sentence.

*"Line 257: 'July 2000-2015'. Should be rewritten to clarify it is for July during those years.*

Done.

*Line 283: Can you show these differences with respect to the ECWMF winds somehow?*

Figure 1a shows the ECMWF monthly mean winds as black arrows and the in situ ship measurements as blue arrows. Thus, we think differences are directly visible in the figure. See also our answer to your next comment.

*Can you add the time varying ECMWF winds in Figure 2, if there is a discrepancy to discuss?*

The comparison of time varying ERA-Interim winds and in situ ship winds can be found in Fig. S1 in the supporting material. In line 285, we added the sentence: "Surface winds from in situ ship measurements, radiosondes, and time varying ERA-Interim data show good agreement (Fig. S1)."

*Line 291: Give the lifetime of butane so the reader can judge what this is testing.*

The atmospheric lifetime of butane is estimated to 2.5 days considering the reaction with OH and ozone (Finlayson-Pitts and Pitts, 2000). We added to line 291: "(lifetime 2.5 days; Finlayson-Pitts and Pitts, 2000)".

*Line 301: Change 'lower' to 'smaller'. (There are many other places where I would suggest changing higher to larger, when higher can be confused with meaning higher altitude).*

Yes, we agree with the reviewer and changed the words "lower" and "higher" in several places in Sect. 3.2 and 3.3 to "smaller" and "larger/stronger", respectively.

*Section 3.3.: This section compares the emission values between the cruises, but I think it is missing an overall synthesis or discussion about what these differences tell us about the different regimes or techniques. I would suggest adding a paragraph after line 389.*

We added the following paragraph at line 389: "In general, the emissions measured during the OASIS cruise in the subtropical and tropical West Indian Ocean were as large as or larger than in other tropical open ocean cruises or studies. Especially $CH_2Br_2$ emissions during the OASIS cruise were larger than any previous emission estimates. The West Indian Ocean seems to be a region with significant contribution to the global open ocean VSLS emissions, especially in boreal summer when wind speeds are high because of the southwest monsoon circulation."

*Line 398: It would be interesting to see some plot of the vertical distribution of tracers in the different transport regimes.*

We show the vertical distribution of bromoform from the OASIS cruise for the four transport regimes averaged over the time of the model calculation in the below Fig. R2. These profiles confirm that the bromoform emissions in the Westerlies and Transitions regime stay mostly below 5 km height, while in the Monsoon Circulation and Local Convection regimes bromoform is more uniformly distributed in the troposphere. We added this figure to the supporting material and explanatory text in line 404: "The different uplift heights are reflected in the vertical distribution of bromoform in each transport regime (Fig. S2)."

[Figure]

**Figure R2: Time-averaged vertical distribution of bromoform in the four transport regimes (Figure 3b).**

*Line 413: I don't understand the definition of transit half-life. It might be the use of 'has reached 17km.' All entrained tracer reaches 17km? It seems like Figure 5 would help but that comes later. For Figure 5 can you add a symbol on the blue line at the half-life value? (If I have understood correctly).*

We used the term "entrained tracer" as a synonym for "has reached 17 km". We changed the explanation to: "…transit half-life, which is the time after which half of the total amount of entrained tracers have been entrained above 17 km altitude." We also added a half-life marker in Fig. 5.

*Line 416: Table 4: Please check all the values in Table 4. I tried to check my understanding of Transport Efficiency by dividing entrainment by emission. E.g. for CHBr3 cruise mean: 5.5/430 =*

*1.28%. Not 1.38. I tried other values and there seemed to be differences (23.6/430 = 5.49% and not 6.38%). What is wrong? Also, it would help if the text used the same precision as the table (e.g. line 419 say 1.38% and not 1.3%, or is it 1.28%?).*

Thanks for checking Table 4 carefully! Your understanding of the transport efficiency is correct. There were slips in Table 4, because of a former version of the manuscript. We recalculated transport efficiencies, corrected the tables and adjusted the precision in the table and Sect. 3.4.

*Figure 5: What are the units of the left-hand y axis?*

The unit on this axis is number. We added the unit to the y-axis description of Fig. 5.

**References**
Finlayson-Pitts, B., and Pitts, J.: Chemistry of the upper and lower atmosphere: Theory, experiments and applications, Academic, US, 2000.

**Changes made to the manuscript**

- Table 4 was corrected.
- Fig. 5 was updated.
- We moved the Local Convection area in Fig. 6. This lead to changes in Fig.6, Table 6, and Sect. 4.2.
- Figure R2 was added to the supporting material.
- Minor changes were made according to the Answers to the Reviewers.

**Marked-up version of the manuscript**

[revised manuscript text omitted]